# Generalization through Discrepancy: Leveraging Distributional Fitting Gaps for AI-Generated Image Detection

## Abstract

The generalization of detectors for AI-generated images remains a critical challenge, as methods trained on one generative family often fail when tested on unseen architectures. To tackle this generalization challenge, we dive into the inherent distribution approximation nature of generative modeling and posit that a universal forensic signal lies in the discrepancy between mathematically precise image rescaling traces and the imperfect approximations learned by generative models through training data. We introduce a novel contrastive pre-training framework that sensitizes a feature extractor to these subtle rescaling artifacts by leveraging their inherent periodic patterns and position shift properties, using only real images for training. Our method sets a new state-of-the-art on both GAN and diffusion-generated benchmarks, validating the efficacy of our method. We introduce a new and robust perspective on detection generalization through the lens of distributional fitting divergence. The code and models will be made publicly available.

## 1 Introduction

The emergence of high-fidelity generative models, from Generative Adversarial Networks (GANs) (Goodfellow et al., 2014) to modern diffusion processes (Ho et al., 2020), has enabled the synthesis of images possessing unprecedented realism and diversity. As these synthetic contents become increasingly indistinguishable from real photographs, they not only foster creativity across artistic and commercial fields but also introduce profound threats to digital trust and authenticity. The unchecked spread of AI-generated forgeries exacerbates issues such as large-scale disinformation and the undermining of evidentiary integrity in legal and journalistic contexts. These challenges highlight the critical demand for reliable detection methods in the broader effort to secure AI-generated media.

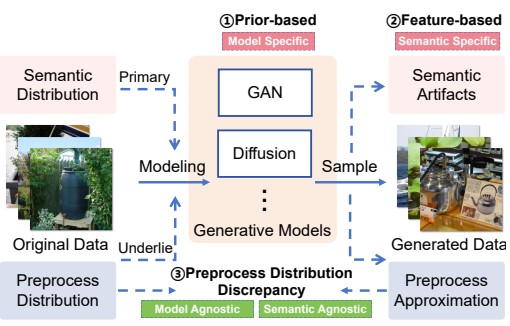

Figure 1: Instead of learning model-specific or semantic-specific features, our framework focuses on the discrepancy of approximated preprocess distribution.

Although detection methods achieve high accuracy in in-domain detection tasks, they often suffer from substantial performance degradation when exposed to samples from unseen generative models. Current research on improving detection generalization largely follows two paradigms: prior-based methods and feature-based methods. Prior-based approaches leverage explicit prior hypotheses to isolate discriminative forensic traces. For instance, Tan et al. (2023) distinguished generated images from real ones in gradient feature space through gradient mapping, while Tan et al. (2024b) enhanced detection by analyzing the upsampling process in generative models and amplifying its artifacts via residual learning. Feature-based methods leverage the powerful feature extraction capabilities of large pre-trained models to enhance detection generalization. For instance, several studies

(Ojha et al., 2023; Tan et al., 2025; Cozzolino et al., 2024; Khan & Dang-Nguyen, 2024) utilize the representational power of large pre-trained models to extract image features, thereby amplifying the artifacts in generated images and improving generalization performance.

Although methods based on prior assumptions perform well on specific generative models, their detection performance degrades significantly when confronted with unseen models that violate those assumptions. For instance, Durall et al. (2020) detected images by leveraging high-frequency anomalies caused by upsampling operations in GANs, which generalizes well across various GAN variants but fails against diffusion-generated samples. Similarly, Wang et al. (2023) discriminated images by analyzing reconstruction residuals specific to the diffusion process, yet exhibits limited effectiveness when applied to GAN-generated content. Such performance drops stem from the inherent limitation of prior-based approaches: observations designed for one class of generative models often do not transfer effectively to others. Recent methods avoid manual prior engineering by leveraging the feature extraction capabilities of large-scale pre-trained models. However, since these pre-trained models are often optimized for high-level semantic tasks (Radford et al., 2021) rather than forensic detection, they tend to overlook critical low-level or high-frequency artifacts, thereby impairing generalization performance (Ojha et al., 2023). Although techniques such as orthogonal decomposition (Yan et al., 2025) attempted to disentangle semantic and forensic features, they still fall short of fully addressing this fundamental mismatch. Therefore, improving detection generalization necessitates solving two core challenges: broadening the coverage of prior-based assumptions to encompass diverse generative classes, and aligning pre-trained feature representations with semantically agnostic, forensically relevant cues.

Towards more generalizable prior assumption, we take inspiration from the intrinsic nature of data distribution fitting in generative models. Generative models approximate the semantic distribution of the given training data, yet inevitably introduce discernible discrepancies in the process. Most existing detection methods focus on this semantic-level discrepancies, such as unnatural texture smoothness (Chen & Yashtini, 2024) or anomalies in high-frequency details (Tan et al., 2024b). However, such semantic discrepancies are highly dependent on the training data distribution and exhibit distinct artifact patterns. Moreover, as generative models continue to evolve, semantic-level discrepancies will gradually diminish, ultimately rendering detection methods that rely solely on such features ineffective. Besides semantic distribution approximations, as indicated by Corvi et al. (2023a), generative modeling also captures underlying distributional characteristics of the training data. For instance, when a generative model is trained on JPEG-compressed images, it tends to produce samples exhibiting similar compression artifacts. This phenomenon is consistently observed across diverse generative frameworks, including both GANs and diffusion models. These underlying distributions include JPEG compression, rescaling operation, and other image processing transformations. Since such processing operations possess rigorous mathematical formulations, the inevitable approximation errors introduced by generative models when fitting these processes create measurable discrepancies from the true mathematical distribution.

In this work, we propose to leverage the discrepancies in generative models' fitting of the underlying distributions. Among these, rescaling stands out as the most widely adopted preprocessing step in generative model pipelines. The prevalence of rescaling in generative training stems from two primary factors: 1) widely-used training datasets are web-sourced like ImageNet (Russakovsky et al., 2015) and LSUN (Yu et al., 2015) which inherently undergo rescaling during collection; 2) model constraints (e.g., fully-connected layers, batching) require fixed input sizes, as Stable Diffusion (Rombach et al., 2022) initially resizes images to $256 \times 256$ before upscaling to $512 \times 512$. Consequently, these rescaling distributions become intrinsically embedded within the generated images. We therefore select rescaling distribution discrepancy as the primary entry point for generalization detection. Crucially, this discrepancy remains largely invariant to variations in semantic content across datasets, thereby offering stronger generalization for detection tasks. Through further analysis of rescaling distribution properties, we construct a contrastive learning task sensitive to these characteristics to pre-train the feature extractor. The trained extractor becomes highly responsive to rescaling distribution discrepancies, enabling feature extraction from a non-semantic perspective. This approach amplifies fitting divergences in generated images, thereby significantly improving generalization in detection tasks. Our main contributions are as follows:

- We are the first to identify and exploit the pre-process distributional discrepancies between real and generated data, which are independent from specific generative architectures, increasing generalization inherently.

- We reveal the periodic patterns and position shift properties of the rescaling operation, which derives a novel semantic-agnostic self-supervised contrastive pre-training task to extract authenticity-oriented features, further enhancing our generalization capability.

- We establish new state-of-the-art on comprehensive benchmarks, outperforming recent methods by a substantial margin. Extensive experiments in few-shot fine-tuning scenarios indicate that the pre-trained feature extractor effectively focuses on authenticity-relevant characteristics.

## 2 RELATED WORK

### 2.1 GENERATIVE MODELS

Generative models have evolved beyond the capabilities of classical autoencoders (Masci et al., 2011; Vincent et al., 2008; Salah et al., 2011), primarily through their capacity to synthesize novel data instances that reflect the true underlying distribution. Although GANs (Goodfellow et al., 2014) once set the standard for image generation with variants such as ProGAN (Karras et al., 2017) for multi-scale learning, StyleGAN (Karras et al., 2019) for style-based control, BigGAN (Brock et al., 2018) for high-capacity synthesis, and StarGAN (Choi et al., 2018) for cross-domain adaptation, they suffer from persistent limitations in output fidelity. Despite advantages in speed and flexibility, GANs frequently introduce perceptible semantic distortions that undermine their practical utility. A paradigm shift occurred with the introduction of diffusion models (Ho et al., 2020; Song et al., 2020a;b), which offer a principled probabilistic framework that not only stabilizes training but also significantly enhances output diversity and coherence. Subsequent large-scale implementations (Rombach et al., 2022; Ramesh et al., 2022; Saharia et al., 2022) have further established the superiority of diffusion processes in producing high-resolution, semantically consistent imagery. Consequently, the marked reduction in generative artifacts has rendered many conventional detection mechanisms increasingly inadequate, particularly those reliant on semantic inconsistencies.

### 2.2 GENERALIZATION DETECTION OF AI-GENERATED IMAGE

Generalization methods for detection can be broadly categorized into two paradigms: the first leverages explicit priors assumptions to extract discriminative forensic artifacts. For instance, Wang et al. (2020) employed data augmentation techniques to effectively enhance cross-model detection performance against GAN-generated images. Subsequent analysis by Durall et al. (2020) demonstrated that the upsampling mechanisms inherent in GAN-based generators produce characteristic high-frequency artifacts, which can be reliably identified via spectral-domain analysis. Tan et al. (2023) leveraged gradient-based analysis to reveal distributional discrepancies between generated and real data, enabling detection through classification in gradient feature space. Zheng et al. (2024) utilized patch-shuffling method to reduce the influence of semantic biases. Liu et al. (2020) and Shiohara & Yamasaki (2022) leveraged the characteristics of underlying textures and features. Although methods based on prior assumptions perform effectively on specific models, their detection efficacy diminishes significantly as new generative architectures emerge, since such priors may not consistently generalize across model architectures.

The second approach utilizes the remarkable feature extraction capability of large-scale pre-trained models for generalization detection. Ojha et al. (2023) first proposed the use of the pre-trained CLIP (Radford et al., 2021) model for generalizable detection of generated images. Since CLIP is trained on large-scale real-world data, it extract features effectively for discrimination. Furthermore, Cozzolino et al. (2024) and Tan et al. (2025) incorporated the textual module into the visual branch of the CLIP framework, capitalizing on its multimodal nature to further enhance the generalization capability of pre-trained models in detecting generated data. However, pre-trained models tend to prioritize high-level semantic information, often lacking in low-level or high-frequency features that are critical for generated detection. Although methods such as Yan et al. (2025) attempted to disentangle semantic and detection-relevant features via orthogonal decomposition, this limitation still adversely affects detection performance. In this work, we explicitly modeling the inherent characteristics of generative models through pre-training on real data, effectively addressing the generalization challenges across diverse generative models.

## 3 PRELIMINARY

In this section, we provide a detailed analysis of rescaling distribution properties. Our examination reveals the rigorous mathematical foundation underlying rescaling operation, which motivates the proposed contrastive learning framework. The rescaling process can be implemented through various interpolation methods. In this work, we focus on bilinear interpolation to elucidate the underlying mechanisms, noting that other interpolation techniques exhibit analogous properties.

**Rescaling via Bilinear Interpolation.** The bilinear interpolation operation consists of two primary stages: 1) mapping of the pixel position, and 2) interpolation of pixel values. The mapping process involves both row and column dimensions and is defined as follows:

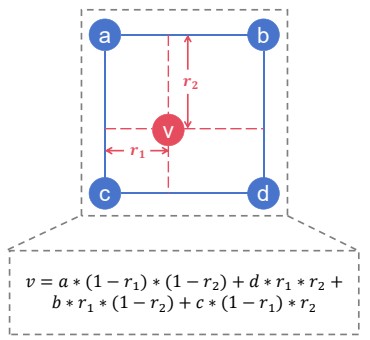

$$x_{\mathrm{src}} = (x_{\mathrm{dst}} + 0.5) \cdot \frac{w_{\mathrm{src}}}{w_{\mathrm{dst}}} - 0.5$$

$$y_{\mathrm{src}} = (y_{\mathrm{dst}} + 0.5) \cdot \frac{h_{\mathrm{src}}}{h_{\mathrm{dst}}} - 0.5 \tag{1}$$

where $(x_{\mathrm{src}}, y_{\mathrm{src}})$ denote the corresponding coordinates in the original source image for a pixel located at $(x_{\mathrm{dst}}, y_{\mathrm{dst}})$ in the rescaled image. $(w_{\mathrm{src}}, h_{\mathrm{src}})$ and $(w_{\mathrm{dst}}, h_{\mathrm{dst}})$ represent the width and height of the original image and rescaled image, respectively.

Figure 2: The interpolation relationship between neighboring pixels in bilinear interpolation method.

Based on the results of the pixel position mapping, the subsequent step involves interpolating the pixel values. The bilinear interpolation process computes the value for a mapped pixel in the source image by performing linear interpolation using its four nearest neighboring pixels, weighted by their relative positional relationships. As illustrated in Figure 2, $r_1$ and $r_2$ represent the relative positional relationships between the mapped pixel and its four nearest neighbors in the original image. They are computed as the fractional parts of the mapped coordinates, obtained by:

$$r_1 = x_{\mathrm{src}} - \lfloor x_{\mathrm{src}} \rfloor \qquad r_2 = y_{\mathrm{src}} - \lfloor y_{\mathrm{src}} \rfloor, \tag{2}$$

where $\lfloor \cdot \rfloor$ denotes the floor function. These fractional components quantify the relative offsets within the unit pixel cell along the horizontal and vertical directions, respectively. The detailed computational procedure is given by the formula shown in the figure.

**Local Distribution Properties of Bilinear Interpolation.** Owing to the mapping and interpolation relationships of pixel positions, our further analysis reveals two key local distribution properties of bilinear interpolation: 1) the periodic distributions in interpolated pixel relationships, and 2) the local dependency among adjacent interpolated pixels. We illustrate both properties in detail using the example provided in Figure 3 which illustrates a bilinear interpolation diagram from a $6 \times 6$ pixel grid to a $4 \times 4$ grid. The blue regions represent pixels from the original image, while the red regions correspond to pixels in the rescaled image. Numeric values within the circles indicate corresponding pixel intensities. Row and column indices for both grids are annotated along the top and left sides of the diagram.

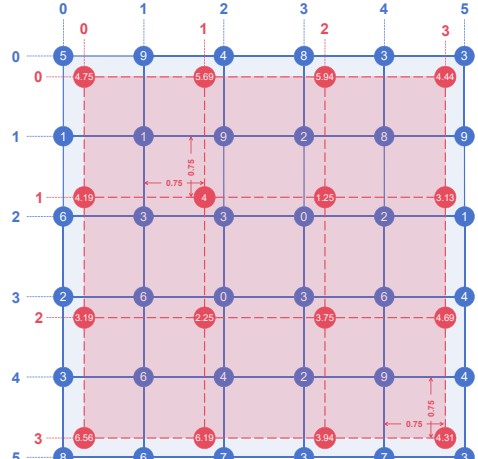

Figure 3: Bilinear interpolation from a $6 \times 6$ (blue) to $4 \times 4$ (red) pixel grid.

From Equation (2), it can be observed that when $x_{\mathrm{src}}$ and $y_{\mathrm{src}}$ are perturbed by integer offsets, the corresponding interpolation ratios $r_1$ and $r_2$ remain unchanged. As a result, the same neighboring pixel interpolation relationships are maintained during the pixel value computation. Returning to Equation (1), when $x_{\mathrm{dst}}$ and $y_{\mathrm{dst}}$ are varied by multiples

of the denominators of $\frac{w_{\text{src}}}{w_{\text{dst}}}$ and $\frac{h_{\text{src}}}{h_{\text{dst}}}$ in their reduced forms, the corresponding changes in $x_{\text{src}}$ and $y_{\text{src}}$ will be integer-valued. Consequently, when $x_{\text{dst}}$ and $y_{\text{dst}}$ vary periodically with a specific period, the resulting interpolation distribution also exhibits periodic behavior. As illustrated in Figure 3 by the points $(1, 1)$ and $(3, 3)$ in the rescaled image, the scaling ratios $\frac{w_{\text{src}}}{w_{\text{dst}}} = \frac{h_{\text{src}}}{h_{\text{dst}}} = \frac{6}{4}$ simplify to $\frac{3}{2}$. Therefore, when $x_{\text{dst}}$ and $y_{\text{dst}}$ are altered with a period of 2, identical interpolation relationships are maintained. Similarly, this property holds for the points $(1, 3)$ and $(3, 1)$.

The second characteristic of bilinear interpolation is its dependency on neighboring pixels during the interpolation process. As shown in Figure 3, the interpolation of both column 0 and column 1 in the rescaled image depends on the pixel values from column 1 of the original image. In contrast, the interpolation of column 1 and column 2 in the rescaled image exhibits no connection and they are entirely independent. This phenomenon arises from accumulated deviations during pixel position mapping, leading to the position shift. The interval of these shifts are determined by the rescaling ratio between original and rescaled images. Such variations in local dependency constitute one of the distinctive features of rescaling distributions under different ratios.

## 4 METHODOLOGY

Through a detailed analysis of the rescaling process, we identify characteristic properties of interpolation distributions. Based on this, we leverage contrastive learning to enable classifiers to model these interpolation distributions, thereby achieving generalized detection of generated images from the perspective of distributional fitting discrepancies.

### 4.1 PROBLEM DEFINITION

Generalized detection of generated images is a binary classification problem aimed at determining whether a given image is model-generated. Typically, the task is formulated as fine-tuning a classifier on data synthesized by a single generative model, while generalizing to detect images from diverse unseen models (Wang et al., 2020). This cross-model generalization encompasses both variants within the same model family (e.g., different GAN architectures) and transfers across distinct families (e.g., from GANs to diffusion models).

Concretely, let $\mathbf{x} \in \mathbb{R}^{h \times w \times 3}$ denotes an RGB input image with height $h$ and width $w$. The source label $y$ of $\mathbf{x}$ belongs to the set $\mathcal{Y} = \{R_1, G_1\}$. A classifier $f_\theta$ is trained on the dataset $\mathcal{D}_{\text{train}} = (\mathbf{x}_i, y_i)$ where $y_i \in \mathcal{Y}$. The trained classifier is then required to generalize to test images $\mathbf{x}'$ drawn from unseen sources $\{R_2, R_3, \ldots, R_N, G_2, G_3, \ldots, G_N\}$, and make predictions according to:

$$\hat{y} = \begin{cases} \text{real}, & \text{if } f_\theta(\mathbf{x}') \geq \tau \\ \text{fake}, & \text{otherwise} \end{cases} \tag{3}$$

where $\tau$ is a decision threshold. Here, $\{G_i\}_{i=1}^N$ denote $N$ distinct generative models, and $\{R_i\}_{i=1}^N$ denote $N$ different real sources.

### 4.2 RESCALING-CONTRASTIVE PRE-TRAINING FOR GENERALIZED DETECTION

**Pre-training via Rescaling Contrastive Learning.** The pre-training process of the classifier consists of three main stages: 1) applying bilinear interpolation to images according to given rescaling ratios, thereby generating images with diverse rescaling distributions; 2) selecting image patches to form positive and negative sample pairs based on their rescaling distributions and relative positional relationships; and 3) pre-training the model on the constructed sample pairs using supervised contrastive learning. The detailed descriptions of the main stages of the pre-training procedure are:

During the image rescaling stage, we randomly select scaling ratios $s \in (1, 2)$, and rescaling each image according to its assigned ratio, as illustrated in Figure 4(a). Here, $s_1$ and $s_2$ denote distinct scaling factors. Although input images may have different original resolutions, those rescaled with the same ratio exhibit identical local rescaling distributions. Due to the pixel position mapping and local interpolation dependencies as analyzed in Section 3, patches from images rescaled with different ratios exhibit distinct interpolation distributions at any location.

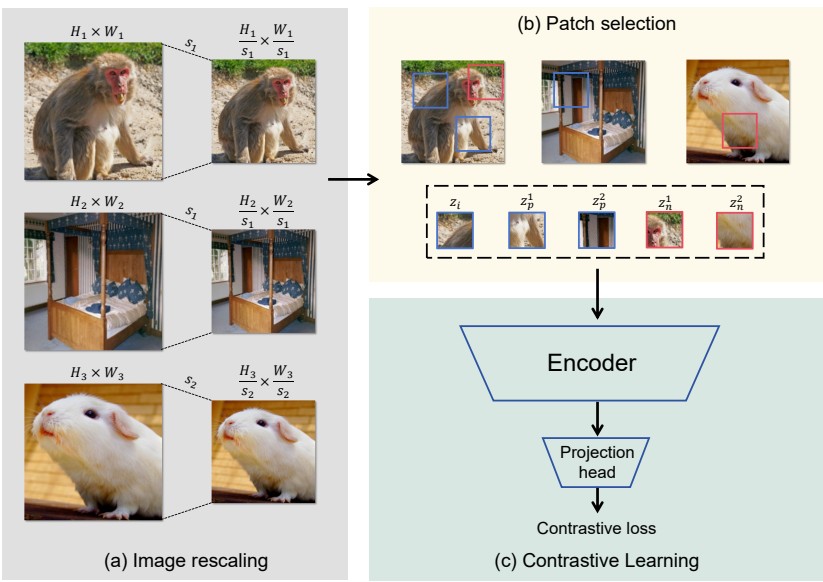

Figure 4: **Pipeline for constructing a contrastive learning setting via image rescaling.** (a) Image rescaling process using different ratio combinations. (b) Selection of positive and negative samples based on rescaling ratios and positional relationships. (c) Pre-training a classification model with contrastive learning to enhance sensitivity to rescaling distributions.

After rescaling different images with assigned ratios, the periodic property of rescaling distributions necessitates that the selection of positive sample pairs must consider both the rescaling ratio and the relative positional relationships of patches. Positive pairs are constructed by selecting patches from images with identical rescaling ratios according to periodic positional relationships. Negative samples comprise two types: 1) patches from images with different rescaling ratios (regardless of position), and 2) patches from images with the same rescaling ratio but at aperiodic positions.

As shown in Figure 4(b), Image 1 and Image 2 share the same rescaling ratio $s_1$, while Image 3 has a different ratio $s_2$. For an anchor patch $z_i$ in Image 1, we compute its periodic positional relationship based on ratio $s_1$, and randomly select another patch $z_p^1$ at a position corresponding to an integer multiple of this period. Similarly, a patch $z_p^2$ is selected from Image 2 following the same periodic relationship. Negative samples are constructed by selecting $z_n^1$ from aperiodic positions in Image 1 and $z_n^2$ from an arbitrary position in Image 3.

We employ a contrastive learning approach to pre-train the model for rescaling distribution modeling, with the details shown in Figure 4(c). The positive and negative sample pairs obtained during the patch selection stage are fed into an encoder, followed by a projection head. The projection head is implemented as a simple Multi-Layer Perceptron (MLP), which maps the encoded features to a latent space where the supervised contrastive loss (Khosla et al., 2020) is computed. The objective function is defined as follows:

$$\mathcal{L}\text{con} = \sum_{i=1}^{N} \frac{-1}{|P(i)|} \sum_{p \in P(i)} \log \frac{\exp(z_i \cdot z_p / \tau)}{\sum_{j=1}^{N} \mathbf{1}_{[j \neq i]} \exp(z_i \cdot z_j / \tau)}, \tag{4}$$

where $z_i$ denotes the projected feature of the $i$-th sample, $P(i)$ represents the set of indices belonging to the same class as sample $i$ within the batch, $\tau$ is a temperature parameter, and $N$ is the batch size.

**Inference.** During inference, the projection head is discarded and replaced with a binary classification head for fine-tuning. The fine-tuning process can be performed either by freezing the encoder parameters and updating only the classification head, or by jointly optimizing the entire network. Beyond generalization detection, since the features extracted by the encoder, which are grounded in rescaling distributions, already capture the discrepancy between the rescaling distributions approximated by generative models and those derived through mathematical modeling, few-shot fine-tuning proves highly effective. This enables incremental learning with minimal samples when encountering unseen generative models.

Table 1: Cross-GAN performance (ACC./A.P.) comparison on the **Self-Synthesis** dataset (9 GAN variants). **Bold** and underline indicate the best and second-best results, respectively.

| Method | Ref | AttGAN | BEGAN | CramerGAN | InfoMaxGAN | MMDGAN | RelGAN | S3GAN | SNGAN | STGAN | **Mean** |
|---|---|---|---|---|---|---|---|---|---|---|---|
| CNNSpot | CVPR2020 (Wang et al., 2020) | 51.1 / 83.7 | 50.2 / 44.9 | 81.5 / 97.5 | 71.1 / 94.7 | 72.9 / 94.4 | 53.3 / 82.1 | 55.2 / 66.1 | 62.7 / 90.4 | 63.0 / 92.7 | 62.3 / 82.9 |
| Frank | PMLR2020 (Frank et al., 2020) | 65.0 / 74.4 | 39.4 / 39.9 | 31.0 / 36.0 | 41.1 / 41.0 | 38.4 / 40.5 | 69.2 / 96.2 | 69.7 / 81.9 | 48.4 / 47.9 | 25.4 / 34.0 | 47.5 / 54.7 |
| Durall | CVPR2020 (Durall et al., 2020) | 39.9 / 38.2 | 48.2 / 30.9 | 60.9 / 67.2 | 50.1 / 51.7 | 59.5 / 65.5 | 80.0 / 88.2 | 87.3 / 97.0 | 54.8 / 58.9 | 62.1 / 72.5 | 60.3 / 63.3 |
| Patchfor | ECCV2020 (Chai et al., 2020) | 68.0 / 92.9 | 97.1 / **100.0** | 97.8 / **99.9** | 93.6 / 98.2 | 97.9 / **100.0** | 99.6 / **100.0** | 66.8 / 68.1 | **97.6** / 99.8 | 92.7 / 99.8 | 90.1 / 95.4 |
| F3Net | ECCV2020 (Qian et al., 2020) | 85.2 / 94.8 | 87.1 / 97.5 | 89.5 / 99.8 | 67.1 / 83.1 | 73.7 / 99.6 | 98.8 / **100.0** | 65.4 / 70.0 | 51.6 / 93.6 | 60.3 / 99.9 | 75.4 / 93.1 |
| GANDetect | ICIP2022 (Mandelli et al., 2022) | 57.4 / 75.1 | 67.9 / **100.0** | 67.8 / 99.7 | 67.6 / 92.4 | 67.7 / 99.3 | 60.9 / 86.2 | 69.6 / 83.5 | 66.7 / 90.6 | 69.6 / 97.2 | 66.1 / 91.6 |
| LGrad | CVPR2023 (Tan et al., 2023) | 68.6 / 93.8 | 69.9 / 89.2 | 50.3 / 54.0 | 71.1 / 82.0 | 57.5 / 67.3 | 89.1 / 99.1 | 78.5 / 86.0 | 78.0 / 87.4 | 54.8 / 68.0 | 68.6 / 80.8 |
| UnivFD | CVPR2023 (Ojha et al., 2023) | 78.5 / 98.3 | 72.0 / 98.9 | 77.6 / 98.8 | 77.6 / 98.9 | 77.6 / 99.7 | 78.2 / 98.7 | 85.2 / 98.1 | 77.6 / 98.7 | 74.2 / 97.8 | 77.6 / 98.8 |
| NPR | CVPR2024 (Tan et al., 2024b) | 83.0 / 96.2 | 99.0 / 99.8 | **98.7** / 99.0 | 94.5 / 98.3 | **98.6** / 99.0 | 99.6 / **100.0** | 79.0 / 80.0 | 88.8 / 97.4 | **98.0** / **100.0** | 93.2 / 96.6 |
| Ours | - | **98.9** / **100.0** | **100.0** / **100.0** | 96.9 / 99.5 | **96.9** / **99.9** | 96.8 / 99.7 | **99.7** / **100.0** | **94.7** / **98.9** | 93.8 / 98.3 | 97.4 / **100.0** | **97.2** / **99.6** |

## 5 EXPERIMENTS

### 5.1 EXPERIMENT SETUP

**Datasets.** To evaluate the generalization performance of the proposed method in practical scenarios, our dataset encompasses a diverse range of GANs, diffusion models, and various real image sources. To assess cross-model generalization across GAN variants, we follow the setting of the NPR (Tan et al., 2024b): classifier is trained on the ForenSynths (Wang et al., 2020) dataset and evaluated on the Self-Synthesis (Tan et al., 2024a) dataset. The ForenSynths dataset contains 20 semantic categories, though only four (i.e., car, cat, chair, horse) are used during training to maintain consistency with prior works. The Self-Synthesis dataset includes multiple GAN variants such as AttGAN (He et al., 2019) and BEGAN (Berthelot et al., 2017). We further examine generalization capability across different diffusion models following C2P-CLIP (Tan et al., 2025), with validation performed on the GenImage (Zhu et al., 2023) dataset. The training subset consists of data generated by the SDv1.4 model (Rombach et al., 2022), while the test set includes samples from multiple diffusion models (e.g., ADM (Dhariwal & Nichol, 2021), GLIDE (Nichol et al., 2021)) as well as the BigGAN model (Brock et al., 2018). Real images are sourced from the LSUN (Yu et al., 2015) and ImageNet (Russakovsky et al., 2015) datasets.

**Implementation Details.** Our method is implemented utilizing the PyTorch (Paszke et al., 2019) framework with 8 NVIDIA 3090 GPUs. The encoder architecture adopts the Xception (Chollet, 2017) backbone. During training, images are randomly cropped to $128 \times 128$ patches, while center cropping is applied during testing. We use the Adam (Kingma & Ba, 2014) optimizer with an initial learning rate of $2 \times 10^{-4}$. First order moment decay rate and second order moment decay rate are set to 0.9 and 0.999, respectively, and weight decay is set to $2 \times 10^{-4}$. The classifier is pre-trained for 200 epochs with a batch size of 128 on ImageNet dataset. Unless otherwise specified, the entire network is jointly fine-tuned by default in subsequent experiments.

**Evaluation Metrics.** Following existing works (Ojha et al., 2023; Tan et al., 2024b; 2025), we compare the effectiveness of different methods using Accuracy (Acc) and Average Precision (AP). The Acc metric is computed with a fixed threshold of 0.5 across all benchmarks, ensuring a fair and consistent comparison of detection performance.

### 5.2 MAIN RESULTS

**Evaluation on Self-Synthesis Dataset.** Table 1 presents the cross-model generalization accuracy across nine GAN architectures. Following the training setting of NPR (Tan et al., 2024b), all competing methods are fine-tuned using ProGAN data across four semantic categories. Our method significantly outperforms the UnivFD baseline (Ojha et al., 2023) by 19.6% and exceeds the current state-of-the-art NPR by 4.0% in classification accuracy, demonstrating strong generalization capability across diverse GAN architectures.

**Evaluation on GenImage Dataset.** Table 2 summarizes the detection accuracy across multiple methods, including those reported in GenImage (Zhu et al., 2023), C2P-CLIP (Tan et al., 2025), DRCT (Chen et al., 2024) and Effort (Yan et al., 2025). All methods are trained using SDv1.4 in the GenImage dataset. The GenImage benchmark incorporates synthetic images produced by advanced diffusion models, including commercial systems such as MidJourney and WuKong. A

Table 2: Accuracy comparison on the **GenImage** dataset with SDv1.4 as the training dataset. **Bold** and underline denote the best and second-best performance, respectively.

| Methods | Ref | SDv1.4 | SDv1.5 | Midjourney | ADM | GLIDE | Wukong | VQDM | BigGAN | mAcc |
|---|---|---|---|---|---|---|---|---|---|---|
| ResNet-50 | CVPR2016 (He et al., 2016) | 99.9 | 99.7 | 54.9 | 53.5 | 61.9 | 98.2 | 56.6 | 52.0 | 72.1 |
| DeiT-S | ICML2021 (Touvron et al., 2021) | 99.9 | 99.8 | 55.6 | 49.8 | 58.1 | 98.9 | 56.9 | 53.5 | 71.6 |
| Swin-T | ICCV2021 (Liu et al., 2021) | 99.9 | 99.8 | 62.1 | 49.8 | 67.6 | 99.1 | 62.3 | 57.6 | 74.8 |
| CNNSpot | CVPR2020 (Wang et al., 2020) | 96.3 | 95.9 | 52.8 | 50.1 | 39.8 | 78.6 | 53.4 | 46.8 | 64.2 |
| Spec | WIFS2019 (Zhang et al., 2019) | 99.4 | 99.2 | 52.0 | 49.7 | 49.8 | 94.8 | 55.6 | 49.8 | 68.8 |
| F3Net | ECCV2020 (Qian et al., 2020) | 99.9 | 99.9 | 50.1 | 49.9 | 50.0 | 99.9 | 49.9 | 49.9 | 68.7 |
| GramNet | CVPR2020 (Liu et al., 2020) | 99.2 | 99.1 | 54.2 | 50.3 | 54.6 | 98.9 | 50.8 | 51.7 | 69.9 |
| UnivFD | CVPR2023 (Ojha et al., 2023) | 96.4 | 96.2 | 93.9 | 71.9 | 85.4 | 94.3 | 81.6 | 90.5 | 88.8 |
| DIRE | ICCV2023 (Wang et al., 2023) | 100.0 | 99.9 | 50.4 | 52.3 | 67.2 | 100.0 | 50.1 | 50.0 | 71.2 |
| FreqNet | AAAI2024 (Tan et al., 2024a) | 98.8 | 98.6 | 89.6 | 66.8 | 86.5 | 97.3 | 75.8 | 81.4 | 86.8 |
| NPR | CVPR2024 (Tan et al., 2024b) | 98.2 | 98.9 | 81.0 | 76.9 | 89.8 | 96.9 | 84.1 | 84.2 | 88.6 |
| FatFormer | CVPR2024 (Liu et al., 2024) | 100.0 | 99.9 | 92.7 | 75.9 | 88.0 | 99.9 | 98.8 | 55.8 | 88.9 |
| DRCT | ICML2024 (Chen et al., 2024) | 95.0 | 94.4 | 91.5 | 79.4 | 89.2 | 94.7 | 90.0 | 81.7 | 89.5 |
| C2P-CLIP | AAAI2025 (Tan et al., 2025) | 90.9 | 97.9 | 88.2 | 96.4 | 99.0 | 98.8 | 96.5 | 98.7 | 95.8 |
| Effort | ICML2025 (Yan et al., 2025) | 99.8 | 99.8 | 82.4 | 78.7 | 93.3 | 97.4 | 91.7 | 77.6 | 91.1 |
| Ours | - | 99.9 | 99.9 | 92.1 | 94.2 | 98.8 | 99.7 | 99.7 | 99.9 | 98.0 |

Table 3: Evaluation on other setups for the proposed method, including ablation study, linear layer fine-tuning and few-shot fine-tuning tasks.

| Methods | SDv1.4 | SDv1.5 | Midjourney | ADM | GLIDE | Wukong | VQDM | BigGAN | mAcc |
|---|---|---|---|---|---|---|---|---|---|
| ***Ablation Study*** | | | | | | | | | |
| Xception | 93.1 | 91.9 | 65.6 | 54.2 | 73.6 | 88.3 | 61.5 | 64.2 | 74.1 |
| Xception+Pre-training(Ours) | 99.9 | 99.9 | 92.1 | 94.2 | 98.8 | 99.7 | 99.7 | 99.9 | 98.0 |
| ***Linear layer Fine-tuning*** | | | | | | | | | |
| UnivFD(fc) | 96.4 | 96.2 | 93.9 | 71.9 | 85.4 | 94.3 | 81.6 | 90.5 | 88.8 |
| Ours(fc) | 99.1 | 98.9 | 90.8 | 91.4 | 95.6 | 98.5 | 98.1 | 96.5 | 96.1 |
| ***Few-shot Fine-tuning*** | | | | | | | | | |
| UnivFD+4-shot | 96.4 | 97.8 | 94.3 | 88.9 | 96.2 | 96.2 | 90.5 | 97.4 | 94.7 |
| NPR+4-shot | 98.2 | 98.6 | 94.5 | 92.3 | 95.7 | 97.5 | 88.6 | 81.8 | 93.4 |
| Ours+4-shot | 99.9 | 99.9 | 98.7 | 98.9 | 99.3 | 100.0 | 99.8 | 99.8 | 99.5 |
| UnivFD+8-shot | 96.4 | 97.8 | 95.7 | 91.6 | 97.3 | 96.7 | 93.9 | 99.0 | 96.1 |
| NPR+8-shot | 98.2 | 98.7 | 96.7 | 96.1 | 97.2 | 97.9 | 89.1 | 86.3 | 95.0 |
| Ours+8-shot | 99.9 | 99.9 | 99.3 | 99.5 | 99.8 | 100.0 | 100.0 | 100.0 | 99.8 |

notable characteristic of this dataset is the inclusion of high-resolution imagery (e.g., 1024×1024 pixels from MidJourney), whose divergence from conventional resolutions introduces significant resolution bias, further challenging detection robustness. Our approach, based on rescaling distribution discrepancy, achieves a new state-of-the-art average accuracy of 98.0%, surpassing the UnivFD baseline and prior best method C2P-CLIP by margins of 9.2% and 2.2%, respectively. The results demonstrate the efficacy of our proposed method in detecting images generated by diffusion models.

**Proposed Pre-training Improves Baseline Performance.** We evaluate the generalization performance of both the baseline method and our approach incorporating the proposed pre-training strategy, as summarized in Table 3. All models are trained on SDv1.4. The results indicate that the Xception model, similar to the ResNet-50 results reported in Table 2, fails to achieve generalization in the baseline setting. However, when enhanced with our pre-training procedure, it attains strong generalization by effectively leveraging distributional discrepancies.

**Linear Layer Fine-tuning.** We further evaluate the scenario where the pre-trained model parameters are frozen and only the linear classification head is fine-tuned. As shown in Table 3, we compare our method with UnivFD, which also fine-tunes a linear layer on top of a frozen CLIP model. The comparative results demonstrate that our pre-training approach learns features more relevant to detection tasks, leading to superior performance.

**Few-shot Fine-tuning.** To address scenarios where limited samples from unseen models available for adaptation, we further evaluate few-shot incremental learning performance, as reported in Table 3. We compare our approach with UnivFD and NPR, which represent pre-training fine-tuning and prior-based feature extraction paradigms, respectively. All models are first trained on SDv1.4, then fine-tuned with either 4-shot or 8-shot samples from each target model before evaluation. The results

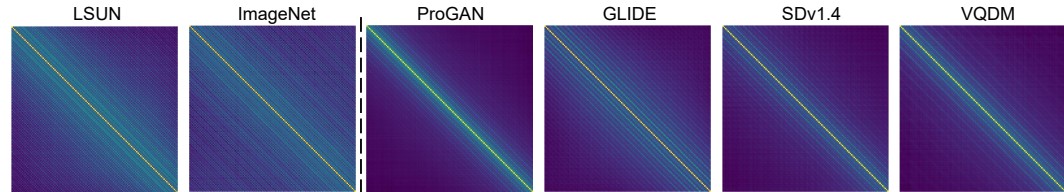

Figure 5: Average cosine similarity map between image patches extracted by the pre-trained model.

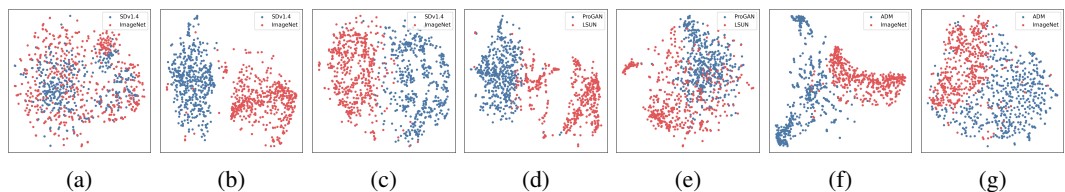

Figure 6: t-SNE visualization of features extracted by different encoders across varying data distributions and post-processing strategies.

show that while UnivFD and NPR exhibit improvements, they struggle to achieve high accuracy on specific categories. In contrast, our method consistently attains superior accuracy across all target models, demonstrating the efficacy of the features learned by our pre-training framework.

**Generated Data Exhibits Distinct Cosine Similarity Map.** To visualize the relationships of the local rescaling distributions within an image, we randomly crop a $256 \times 256$ region from each image sample. Along the diagonal direction, we extract 128 patches of size $128 \times 128$ with a stride of 1. Each patch is processed by our proposed pre-trained model to extract features, resulting in a feature matrix of size $128 \times 2048$, where 2048 is the output dimension of the model. By computing the pairwise cosine similarity between all patch features, we obtain a $128 \times 128$ similarity matrix. This matrix is averaged over 1000 images to produce the final similarity map, as shown in Figure 5. The resulting visualization reveals that real images exhibit multiple bright bands parallel to the diagonal, indicating the periodic nature of local interpolation distributions across different rescaling operations. In contrast, generated images lack such distinctive patterns, revealing a distributional fitting discrepancy which enables effective generalization detection.

**Pre-trained Extractor Remains Robust against Post-rescaling.** Figure 6 presents t-SNE visualizations of features extracted by different models under varying data categories and post-processing conditions. (a) shows features from the CLIP used in UnivFD, which struggles to distinguish generative images. (b)-(g) show the features extracted from our proposed pre-trained model. Specifically, (b), (d) and (f) display evaluations on different generative models using randomly cropped $256 \times 256$ patches from original images. (c), (e) and (g) show corresponding features of images rescaled before feature extracting, where images are rescaled from $256 \times 256$ to $224 \times 224$(c), $192 \times 192$(e), and $160 \times 160$(g), respectively. The results demonstrate that our pre-trained model maintains clear separability across both original and rescaled images, indicating that our approach does not rely merely on the presence or absence of rescaling artifacts. Instead, it operates on more fine-grained distributional discrepancies inherent in the approximations of generative models.

## 6 CONCLUSION

This paper proposes a novel detection method that leverages distributional discrepancies in rescaling operations. By analyzing interpolation properties, we identify consistent fitting gaps between generative models and mathematical rescaling. Our contrastive pre-training framework enables models to learn these fine-grained discrepancies rather than semantic features. Experiments show state-of-the-art performance across GANs and diffusion models, with strong generalization in various settings. Visualizations confirm the method captures fundamental distributional properties beyond superficial artifacts. This work provides a new perspective of distributional discrepancy for generalization detection against evolving generative AI.

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

# A APPENDIX

## A.1 THE USE OF LARGE LANGUAGE MODELS (LLMS)

We employ large language models (LLMs) solely for text polishing and revision of the writing in our paper. Their use is strictly limited to linguistic refinement and does not extend to the methodological contributions, experimental design, or any substantive technical content presented in this work.

## A.2 IMPLEMENTATION DETAILS OF OTHER INTERPOLATION APPROACHES

PyTorch provides several interpolation methods, including NEAREST, BOX, BILINEAR, BICUBIC, LANCZOS, and HAMMING. We have elaborated on the specific procedure of BILINEAR interpolation and identified two key characteristics: periodic distributions and local dependency. In this part, we analyze the detailed mechanisms of other commonly used interpolation methods and demonstrate that, with the exception of NEAREST interpolation, all other methods exhibit these two characteristics. Although NEAREST interpolation does not share these properties, it is not used in data preprocessing of generative models due to the severe artifacts it introduces during rescaling. Thus, NEAREST interpolation is not discussed in this section.

### A.2.1 BOX INTERPOLATION

Given an output pixel at integer coordinates $(x_{\text{dst}}, y_{\text{dst}})$, its value is computed solely from the portion of the input image that maps to it.

**Compute the Input-Space Region of This Output Pixel.** Let the input image size be $(w_{\text{src}}, h_{\text{src}})$ and the output size be $(w_{\text{dst}}, h_{\text{dst}})$. The scaling factors are

$$s_x = \frac{w_{\text{src}}}{w_{\text{dst}}}, \qquad s_y = \frac{h_{\text{src}}}{h_{\text{dst}}}.$$

The continuous region in the input domain corresponding to this single output pixel is

$$R_d = [\, x_{\text{dst}} s_x, \; (x_{\text{dst}} + 1) s_x \,] \times [\, y_{\text{dst}} s_y, \; (y_{\text{dst}} + 1) s_y \,].$$

**Find All Input Pixels Overlapping This Region.** An input pixel at integer coordinates $(x, y)$ spans the unit square

$$P_{xy} = [x, x + 1] \times [y, y + 1].$$

It contributes to the output pixel $(x_{\text{dst}}, y_{\text{dst}})$ if

$$P_{xy} \cap R_d \neq \varnothing.$$

**Compute the Overlap Area.** For each overlapping input pixel $(x, y)$, compute the width and height of the intersection:

$$w_{xy} = \max\Big(0,\ \min(x + 1,\ (x_{\mathrm{dst}} + 1)s_x) - \max(x,\ x_{\mathrm{dst}}s_x)\Big),$$

$$h_{xy} = \max\Big(0,\ \min(y + 1,\ (y_{\mathrm{dst}} + 1)s_y) - \max(y,\ y_{\mathrm{dst}}s_y)\Big).$$

The intersection area is

$$A_{xy} = w_{xy} \cdot h_{xy}.$$

**Compute the Output Pixel Value.** Let $I(x, y)$ be the input pixel value. The BOX interpolation result of this single output pixel is

$$O(x_{\mathrm{dst}}, y_{\mathrm{dst}}) = \frac{1}{s_x s_y} \sum_{\text{overlapping } (x,y)} I(x, y)\, A_{xy}.$$

**Interpolation Characteristics.** In the BOX interpolation technique, each output pixel is first mapped back to the source image according to the scaling factors. The contribution of each source pixel is then determined by its proportional overlap area with the pixel region, which serves as the interpolation weight. Consequently, similar to BILINEAR interpolation, BOX interpolation exhibits both periodicity and local dependency.

### A.2.2 BICUBIC INTERPOLATION

Given an output pixel at integer coordinates $(x_{\mathrm{dst}}, y_{\mathrm{dst}})$, BICUBIC interpolation computes its value by sampling a $4 \times 4$ neighborhood around the corresponding position in the source image.

**Coordinate Mapping.** Let the input image have size $(w_{\mathrm{src}}, h_{\mathrm{src}})$ and the output size be $(w_{\mathrm{dst}}, h_{\mathrm{dst}})$. The scaling factors are

$$s_x = \frac{w_{\mathrm{src}}}{w_{\mathrm{dst}}}, \qquad s_y = \frac{h_{\mathrm{src}}}{h_{\mathrm{dst}}}.$$

The output pixel $(x_{\mathrm{dst}}, y_{\mathrm{dst}})$ corresponds to the input coordinate

$$x_{\mathrm{src}} = (x_{\mathrm{dst}} + 0.5)\, s_x - 0.5, \qquad y_{\mathrm{src}} = (y_{\mathrm{dst}} + 0.5)\, s_y - 0.5.$$

Let

$$x_0 = \lfloor x_s \rfloor, \qquad y_0 = \lfloor y_s \rfloor.$$

**Bicubic Kernel.** The cubic convolution kernel with parameter $a$ (typically $a = -0.5$) is

$$k(t) = \begin{cases} (a + 2)|t|^3 - (a + 3)|t|^2 + 1, & 0 \le |t| < 1, \\ a|t|^3 - 5a|t|^2 + 8a|t| - 4a, & 1 \le |t| < 2, \\ 0, & |t| \ge 2. \end{cases}$$

**Compute Horizontal and Vertical Weights.** For the horizontal direction:

$$w_i = k(x_{\mathrm{src}} - (x_0 + i)), \qquad i \in \{-1, 0, 1, 2\}.$$

For the vertical direction:

$$v_j = k(y_{\mathrm{src}} - (y_0 + j)), \qquad j \in \{-1, 0, 1, 2\}.$$

**Bicubic Combination Over a $4 \times 4$ Neighborhood.** Let $I(x, y)$ denote the source image pixel value (per channel). The bicubic interpolated output pixel is

$$O(x_{\mathrm{dst}}, y_{\mathrm{dst}}) = \sum_{j=-1}^{2} \sum_{i=-1}^{2} I(x_0 + i,\ y_0 + j)\, w_i\, v_j.$$

**Interpolation Characteristics.** In BICUBIC interpolation, the mapping of pixel positions follows the same procedure as in BILINEAR interpolation. The key difference is that each mapped position is reconstructed using a weighted combination of a $4 \times 4$ neighborhood of surrounding pixels. Consequently, BICUBIC interpolation exhibits the same interpolation characteristics as BILINEAR interpolation.

### A.2.3 LANCZOS INTERPOLATION

Given an output pixel at integer coordinates $(x_{\mathrm{dst}}, y_{\mathrm{dst}})$, LANCZOS interpolation reconstructs its value by a separable, windowed-sinc filter with finite support parameter $a$ (commonly $a = 2$ or $a = 3$).

**Coordinate Mapping.** Let the source image size be $(w_{\mathrm{src}}, h_{\mathrm{src}})$ and the destination size be $(w_{\mathrm{dst}}, h_{\mathrm{dst}})$. Define the scaling factors

$$s_x = \frac{w_{\mathrm{src}}}{w_{\mathrm{dst}}}, \qquad s_y = \frac{h_{\mathrm{src}}}{h_{\mathrm{dst}}}.$$

Map the integer destination pixel to a source coordinate using the commonly used center-preserving formula

$$x_{\mathrm{src}} = (x_{\mathrm{dst}} + 0.5)\, s_x - 0.5, \qquad y_{\mathrm{src}} = (y_{\mathrm{dst}} + 0.5)\, s_y - 0.5.$$

Let

$$x_0 = \lfloor x_{\mathrm{src}} \rfloor, \qquad y_0 = \lfloor y_{\mathrm{src}} \rfloor.$$

**Lanczos Kernel.** Define the normalized sinc function

$$\mathrm{sinc}(t) = \begin{cases} \dfrac{\sin(\pi t)}{\pi t}, & t \neq 0, \\ 1, & t = 0. \end{cases}$$

The Lanczos kernel with window parameter $a > 0$ is

$$L_a(t) = \begin{cases} \mathrm{sinc}(t)\, \mathrm{sinc}\!\left(\frac{t}{a}\right), & |t| < a, \\ 0, & |t| \geq a. \end{cases}$$

**Determine the Contributing Source Samples.** The kernel is nonzero only for offsets satisfying $|t| < a$. Thus the integer source indices contributing in the horizontal direction are the set

$$\mathcal{I} = \{\, i \in \mathbb{Z} \ : \ |x_{\mathrm{src}} - i| < a \,\},$$

and in the vertical direction

$$\mathcal{J} = \{\, j \in \mathbb{Z} \ : \ |y_{\mathrm{src}} - j| < a \,\}.$$

Equivalently one can enumerate

$$i \in \{\, x_0 - (a-1), \ \ldots, \ x_0 + a \,\}, \qquad j \in \{\, y_0 - (a-1), \ \ldots, \ y_0 + a \,\}.$$

**Compute Separable Weights.** For each contributing horizontal index $i$ and vertical index $j$, compute the separable weights

$$w_i = L_a(x_{\mathrm{src}} - i), \qquad v_j = L_a(y_{\mathrm{src}} - j).$$

The combined 2D weight for source sample $(i, j)$ is the product

$$W_{ij} = w_i\, v_j.$$

**Compute the Output Pixel Value.** Let $I(i, j)$ denote the source image sample value (per channel). The Lanczos interpolated value for this single destination pixel is the normalized weighted sum over the contributing neighborhood:

$$O(x_{\mathrm{dst}}, y_{\mathrm{dst}}) = \frac{\displaystyle\sum_{j \in \mathcal{J}} \sum_{i \in \mathcal{I}} I(i, j)\, W_{ij}}{\displaystyle\sum_{j \in \mathcal{J}} \sum_{i \in \mathcal{I}} W_{ij}}.$$

**Interpolation Characteristics.** In LANCZOS interpolation, the mapping of pixel positions follows the same procedure as in BILINEAR interpolation. The key difference is that each mapped position is reconstructed using a weighted combination of a $2a \times 2a$ neighborhood of surrounding pixels. Consequently, LANCZOS interpolation exhibits the same interpolation characteristics as BILINEAR interpolation.

### A.2.4 HAMMING INTERPOLATION

HAMMING interpolation is a windowed-sinc interpolation method in which the ideal sinc kernel is multiplied by a Hamming window. For a single destination pixel located at integer coordinates $(x_{\mathrm{dst}}, y_{\mathrm{dst}})$, the value is computed by sampling a finite neighborhood in the source image using this kernel.

**Coordinate Mapping.** Let the source image have size $(w_{\mathrm{src}}, h_{\mathrm{src}})$ and the destination size be $(w_{\mathrm{dst}}, h_{\mathrm{dst}})$. The scaling factors are

$$s_x = \frac{w_{\mathrm{src}}}{w_{\mathrm{dst}}}, \qquad s_y = \frac{h_{\mathrm{src}}}{h_{\mathrm{dst}}}.$$

Map the destination pixel center to a continuous source location:

$$x_{\mathrm{src}} = (x_{\mathrm{dst}} + 0.5)\, s_x - 0.5, \qquad y_{\mathrm{src}} = (y_{\mathrm{dst}} + 0.5)\, s_y - 0.5.$$

Define

$$x_0 = \lfloor x_{\mathrm{src}} \rfloor, \qquad y_0 = \lfloor y_{\mathrm{src}} \rfloor.$$

**Hamming Windowed Sinc Kernel.** Define the normalized sinc:

$$\mathrm{sinc}(t) = \begin{cases} \dfrac{\sin(\pi t)}{\pi t}, & t \neq 0, \\ 1, & t = 0. \end{cases}$$

The Hamming window is

$$w(t) = 0.54 + 0.46 \cos\left( \frac{\pi t}{a} \right), \qquad |t| < a,$$

where $a$ is the window radius (typically $a = 2$).

The Hamming interpolation kernel is the product

$$H_a(t) = \begin{cases} \mathrm{sinc}(t)\, w(t), & |t| < a, \\ 0, & |t| \geq a. \end{cases}$$

**Contributing Source Samples.** The kernel has finite support $|t| < a$. Thus, contributing horizontal and vertical indices are

$$\mathcal{I} = \{\, i \in \mathbb{Z} \mid |x_{\mathrm{src}} - i| < a \,\}, \qquad \mathcal{J} = \{\, j \in \mathbb{Z} \mid |y_{\mathrm{src}} - j| < a \,\}.$$

Equivalently,

$$i \in \{\, x_0 - (a-1), \ldots, x_0 + a \,\}, \quad j \in \{\, y_0 - (a-1), \ldots, y_0 + a \,\}.$$

**Separable Weights.** Compute the 1D horizontal and vertical weights:

$$w_i = H_a(x_{\mathrm{src}} - i), \qquad v_j = H_a(y_{\mathrm{src}} - j).$$

The 2D separable weight is

$$W_{ij} = w_i\, v_j.$$

**Compute the Output Pixel Value.** Let $I(i,j)$ denote the source pixel values. The Hamming-interpolated destination pixel is

$$O(x_{\mathrm{dst}}, y_{\mathrm{dst}}) = \frac{\displaystyle\sum_{j \in \mathcal{J}} \sum_{i \in \mathcal{I}} I(i,j)\, W_{ij}}{\displaystyle\sum_{j \in \mathcal{J}} \sum_{i \in \mathcal{I}} W_{ij}}.$$

Table 4: Accuracy and AUC comparison on the **GenImage** (unbiased) dataset with SDv1.4 as the training dataset. **Bold** and underline denote the best and second-best performance, respectively.

| Methods | SDv1.4 | SDv1.5 | Midjourney | ADM | GLIDE | Wukong | VQDM | BigGAN | **Mean** |
|---|---|---|---|---|---|---|---|---|---|
| CNNDetect(Wang et al., 2020) | 50.1 / 65.1 | 49.9 / 66.4 | 50.1 / 79.3 | 49.9 / 51.8 | 50.7 / 59.4 | 50.2 / 62.6 | 51.2 / 63.4 | 58.4 / 70.9 | 51.3 / 64.8 |
| DMID(Corvi et al., 2023b) | **99.9 / 100.** | **99.8 / 100.** | **97.4 / 100.** | 51.3 / 78.5 | 56.6 / 94.9 | **99.6 / 100.** | 75.1 / 97.6 | 52.3 / 74.6 | 79.0 / 93.2 |
| LGrad(Tan et al., 2023) | 49.8 / 50.0 | 49.1 / 49.2 | 50.6 / 50.5 | 30.5 / 24.6 | 30.0 / 22.2 | 46.9 / 47.6 | 30.8 / 23.9 | 28.9 / 18.7 | 39.6 / 35.8 |
| UnivFD(Ojha et al., 2023) | 55.5 / 78.7 | 56.6 / 78.1 | 54.2 / 74.0 | 64.4 / 85.2 | 63.9 / 88.8 | 63.7 / 86.9 | 79.7 / 94.8 | 86.1 / 96.7 | 65.5 / 85.4 |
| DeFake(Sha et al., 2023) | 85.1 / 93.3 | 85.4 / 93.4 | 79.2 / 87.7 | 48.5 / 49.3 | 80.4 / 87.9 | 81.8 / 89.8 | 64.4 / 71.1 | 64.4 / 72.6 | 73.7 / 80.6 |
| DIRE(Wang et al., 2023) | 47.3 / 41.7 | 47.3 / 39.8 | 47.5 / 38.0 | 46.7 / 25.3 | 47.0 / 29.9 | 47.7 / 45.4 | 47.7 / 35.0 | 46.9 / 26.6 | 47.3 / 35.2 |
| AntifakePrompt(Chang et al., 2023) | 77.1 / - | 76.6 / - | 70.4 / - | 81.6 / - | 81.8 / - | 77.6 / - | 81.1 / - | 81.7 / - | 78.5 / - |
| NPR(Tan et al., 2024b) | 49.4 / 54.3 | 49.7 / 53.3 | 47.4 / 42.3 | 50.5 / 46.9 | 48.3 / 42.1 | 50.2 / 52.4 | 53.9 / 52.3 | 56.3 / 56.9 | 50.7 / 50.1 |
| FatFormer(Liu et al., 2024) | 52.0 / 49.8 | 53.3 / 48.7 | 51.6 / 46.2 | 60.4 / 69.1 | 65.1 / 78.4 | 58.1 / 61.6 | 71.5 / 84.5 | 80.1 / 88.5 | 61.5 / 65.9 |
| FasterThanLies(Lanzino et al., 2024) | 92.2 / 97.8 | 92.3 / 97.9 | 69.7 / 83.1 | 77.2 / 88.6 | 66.1 / 83.0 | 88.1 / 95.4 | 76.6 / 86.8 | 54.1 / 78.9 | 77.0 / 88.9 |
| RINE(Koutlis & Papadopoulos, 2024) | 60.5 / 93.9 | 61.1 / 94.1 | 52.4 / 86.3 | 63.9 / 93.8 | 74.7 / 98.1 | 70.0 / 95.7 | 81.4 / 98.4 | 88.5 / 99.4 | 69.1 / 95.0 |
| AIDE(Yan et al., 2024) | 74.5 / 98.2 | 75.9 / 98.5 | 57.4 / 88.1 | 50.1 / 61.2 | 52.3 / 80.4 | 69.3 / 95.9 | 51.0 / 78.0 | 50.7 / 73.1 | 60.2 / 84.2 |
| LaDeDaCavia et al. (2024) | 54.8 / 55.6 | 53.0 / 53.6 | 52.1 / 51.3 | 34.6 / 6.8 | 34.5 / 8.7 | 57.7 / 61.6 | 34.8 / 10.8 | 80.3 / 93.1 | 50.2 / 42.7 |
| C2P-CLIPTan et al. (2025) | 80.5 / 94.4 | 79.1 / 94.3 | 55.9 / 74.3 | 71.3 / 86.7 | 74.8 / 93.6 | 81.0 / 93.1 | 74.1 / 92.2 | 87.5 / 97.2 | 75.5 / 91.0 |
| CoDE(Baraldi et al., 2024) | 96.6 / 99.4 | 96.5 / 99.2 | 69.6 / 86.0 | 51.9 / 53.7 | 58.0 / 78.1 | 95.0 / 99.1 | 56.0 / 66.8 | 50.0 / 70.2 | 71.7 / 81.6 |
| B-Free(Guillaro et al., 2025) | 98.8 / **100.** | 98.7 / **100.** | 95.7 / 99.2 | 79.8 / 93.0 | 85.3 / 95.8 | 99.0 / **100.** | 88.7 / 97.0 | 68.7 / 94.1 | 89.3 / 97.4 |
| Ours | 99.2 / 99.8 | 98.7 / 99.8 | 96.7 / 99.5 | **98.7 / 99.7** | **98.9 / 99.8** | 98.2 / 99.6 | **98.5 / 99.7** | **98.7 / 99.8** | **98.4 / 99.7** |

Table 5: Average precision comparison on GANs from the **UniversalFakeDetect** dataset with Pro-GAN as the training dataset.

| Methods | ProGAN | StyleGAN | StyleGAN2 | BigGAN | CycleGAN | StarGAN | GauGAN | Deepfake | **Mean** |
|---|---|---|---|---|---|---|---|---|---|
| CNNDet(Wang et al., 2020) | 99.2 | 91.4 | 96.7 | 73.3 | 88.2 | 90.7 | 92.2 | 62.3 | 86.7 |
| FreDect(Frank et al., 2020) | 85.2 | 72.2 | 71.4 | 86.5 | 71.7 | 99.5 | 77.4 | 49.2 | 76.6 |
| LGrad(Tan et al., 2023) | 100.0 | 99.9 | 99.9 | 90.5 | 94.7 | 100.0 | 79.2 | 67.8 | 91.6 |
| UFD(Ojha et al., 2023) | 100.0 | 98.8 | 98.6 | 99.1 | 99.6 | 100.0 | 99.2 | 90.2 | 98.2 |
| PatchCraft(Zhong et al., 2023b) | 100.0 | 98.7 | 97.7 | 99.3 | 85.1 | 100.0 | 81.8 | 79.6 | 92.7 |
| FreqNet(Tan et al., 2024a) | 100.0 | 99.6 | 95.5 | 95.5 | 99.7 | 100.0 | 98.6 | 94.5 | 97.9 |
| NPR(Tan et al., 2024b) | 100.0 | 100.0 | 100.0 | 94.5 | 95.7 | 100.0 | 88.2 | 86.1 | 95.6 |
| FatFormer(Liu et al., 2024) | 100.0 | 99.6 | 99.8 | 100.0 | 99.8 | 100.0 | 100.0 | 97.6 | 99.5 |
| SAFE(Li et al., 2025) | 100.0 | 99.8 | 100.0 | 95.4 | 99.8 | 100.0 | 97.0 | 97.5 | 98.7 |
| CoD(Jia et al., 2025) | 100.0 | 99.9 | 99.9 | 98.0 | 99.9 | 100.0 | 99.9 | 98.5 | 99.6 |
| Ours | 100.0 | 100.0 | 100.0 | 99.3 | 99.2 | 100.0 | 98.6 | 91.7 | 98.6 |

**Interpolation Characteristics.** In HAMMING interpolation, the mapping of pixel positions follows the same procedure as in BILINEAR interpolation. The key difference is that each mapped position is reconstructed using a weighted combination of a $2(a-1) \times 2(a-1)$ neighborhood of surrounding pixels. Consequently, HAMMING interpolation exhibits the same interpolation characteristics as BILINEAR interpolation.

## A.3 ADDITIONAL GENERALIZATION ANALYSIS

**Evaluation on GenImage (unbiased) Dataset.** Table 4 presents the evaluation results on the GenImage (unbiased) dataset. Following the experimental settings in Guillaro et al. (2025) and Grommelt et al. (2024), we test the classifier's detection performance on JPEG-compressed generated images from the GenImage dataset. Our method significantly outperforms current state-of-the-art approaches, achieving an average accuracy improvement of 9.1% over the B-Free method. This demonstrates that our approach does not rely on JPEG compression artifacts as shortcuts during detection.

**Evaluation on UniversalFakeDetect Dataset.** Tables 5 and 6 present the evaluation results on GAN and Diffusion models from the UniversalFakeDetect dataset, respectively. Following the experimental settings in Ojha et al. (2023) and Jia et al. (2025), the model is trained on ProGAN-generated data and tested on various other models. The results demonstrate that our method achieves competitive performance across both GAN and Diffusion models, maintaining consistency with current state-of-the-art results.

**Evaluation on Synthbuster Dataset.** Table 7 presents the evaluation results on the Synthbuster dataset. Following the experimental settings in Bammey (2023) and Karageorgiou et al. (2025), we test our method on various high-resolution, high-quality diffusion model samples. The experimental results demonstrate that our approach remains effective in detecting images generated by high-quality diffusion models. Moreover, the detection performance on original images without post-processing remains robust, as our method maintains the capability to capture the rescaling dis-

Table 6: Average precision comparison on Diffusions from the **UniversalFakeDetect** dataset with ProGAN as the training dataset.

| Methods | DALL-E | Glide_100_10 | Glide_100_27 | Glide_50_27 | ADM | LDM_100 | LDM_200 | LDM_200_cfg | **Mean** |
|---|---|---|---|---|---|---|---|---|---|
| CNNDet(Wang et al., 2020) | 61.2 | 72.9 | 71.3 | 76.1 | 66.6 | 63.7 | 64.5 | 63.1 | 67.5 |
| FreDect(Frank et al., 2020) | 62.5 | 44.3 | 40.8 | 42.3 | 52.5 | 51.3 | 50.9 | 52.4 | 49.6 |
| LGrad(Tan et al., 2023) | 97.3 | 94.9 | 93.2 | 95.0 | 99.8 | 99.2 | 99.1 | 99.2 | 97.3 |
| UFD(Ojha et al., 2023) | 96.5 | 96.5 | 97.0 | 97.2 | 84.5 | 97.0 | 97.0 | 88.6 | 94.3 |
| PatchCraft(Zhong et al., 2023b) | 93.0 | 92.0 | 93.9 | 88.7 | 90.5 | 97.7 | 97.9 | 96.9 | 93.8 |
| FreqNet(Tan et al., 2024a) | 99.5 | 96.1 | 96.6 | 95.0 | 74.5 | 99.6 | 99.0 | 99.0 | 94.9 |
| NPR(Tan et al., 2024b) | 99.5 | 99.8 | 99.7 | 99.8 | 81.0 | 99.0 | 99.9 | 99.9 | 97.4 |
| FatFormer(Liu et al., 2024) | 99.8 | 99.5 | 99.3 | 99.1 | 91.8 | 99.8 | 99.8 | 99.0 | 98.4 |
| SAFE(Li et al., 2025) | 99.7 | 99.4 | 98.9 | 99.2 | 95.7 | 100.0 | 100.0 | 99.8 | 99.0 |
| CoD(Jia et al., 2025) | 99.6 | 99.6 | 99.5 | 99.5 | 97.4 | 99.8 | 100.0 | 99.8 | 99.4 |
| Ours | 100.0 | 99.3 | 98.6 | 98.8 | 99.9 | 100.0 | 100.0 | 99.9 | 99.6 |

Table 7: AUC comparison on the **Synthbuster** dataset. **Bold** and underline denote the best and second-best performance, respectively.

| Methods | Glide | SD1.3 | SD1.4 | SD2 | SDXL | MJv5 | DALLE2 | DALLE3 | Firefly | **Mean** |
|---|---|---|---|---|---|---|---|---|---|---|
| NPR(Tan et al., 2024b) | 72.2 | 89.6 | 60.5 | 12.5 | 18.1 | 15.3 | 3.9 | **97.1** | 38.0 | 45.2 |
| Dire(Wang et al., 2023) | 33.3 | 59.9 | 68.5 | 61.9 | 46.9 | 41.9 | 52.2 | 65.2 | 49.9 | 53.2 |
| CNNDet(Wang et al., 2020) | 59.2 | 59.0 | 61.2 | 57.5 | 67.4 | 48.8 | 71.5 | 23.5 | 73.4 | 57.9 |
| FreqNet(Tan et al., 2024a) | 43.6 | 92.3 | 92.7 | 42.5 | 66.5 | 36.9 | 47.4 | 42.2 | 80.9 | 60.6 |
| Fusing(Ju et al., 2022) | 63.0 | 62.8 | 62.2 | 66.9 | 62.1 | 64.0 | 76.7 | 25.2 | 76.3 | 62.1 |
| LGrad(Tan et al., 2023) | 76.5 | 82.4 | 83.4 | 60.7 | 70.2 | 69.2 | 85.7 | 30.0 | 42.0 | 66.7 |
| UnivFD(Ojha et al., 2023) | 63.3 | 80.8 | 81.2 | 84.3 | 78.3 | 57.1 | 91.4 | 31.0 | 95.5 | 73.7 |
| GramNet(Liu et al., 2020) | 78.2 | 83.9 | 84.3 | 66.7 | 77.8 | 63.8 | 85.2 | 42.9 | 38.0 | 69.0 |
| DeFake(Sha et al., 2023) | 86.1 | 64.2 | 63.6 | 66.2 | 52.3 | 67.0 | 41.4 | 93.3 | 39.4 | 63.7 |
| PatchCr(Zhong et al., 2023a) | 78.4 | 95.7 | 96.2 | 95.7 | 96.7 | 71.8 | 81.8 | 28.1 | 79.1 | 81.2 |
| DMID(Corvi et al., 2023b) | 73.1 | **100.0** | **100.0** | 99.7 | 99.6 | 99.9 | 54.3 | 41.3 | 90.2 | 84.2 |
| RINE(Koutlis & Papadopoulos, 2024) | **95.6** | **99.9** | 99.9 | 96.6 | 99.3 | 96.4 | 93.0 | 41.8 | 82.9 | 89.5 |
| SPAI(Karageorgiou et al., 2025) | 90.2 | 99.6 | 99.6 | 96.5 | 97.4 | 94.5 | 91.1 | 90.2 | **96.0** | 95.0 |
| Ours | 92.1 | 99.7 | 99.7 | **100.0** | **100.0** | **100.0** | 97.1 | 93.0 | 91.5 | **97.0** |

tribution characteristics from the source data. Consequently, our approach surpasses current state-of-the-art methods.

**Robustness to Perturbations.** In addition to evaluating on clean images, we also evaluate the classifier's detection capability under various image degradation scenarios. In real-world applications, images may undergo multiple perturbations during propagation, making robust detection under degraded conditions crucial for practical deployment. Following prior works Wang et al. (2020) and Ojha et al. (2023), we test four types of perturbations: Rescaling (with scaling factor $\alpha$), JPEG compression (with quality parameter $q$), Gaussian blur (with standard deviation $\sigma$), and Gaussian noise (with standard deviation $\sigma$). As shown in Figure 7, our method maintains strong detection performance across different perturbation scenarios. Although our approach relies on rescaling distributions, its fine-grained pretraining enables the model to remain unaffected by rescaling post-processing while still effectively discerning differences in image rescaling distributions.

**Consistent Performance under Patch Size Variations.** Table 8 presents the generalization results of the proposed method under different patch sizes, with experimental settings consistent with Table 2. The model was pre-trained using a patch size of 128, while fine-tuning and testing were

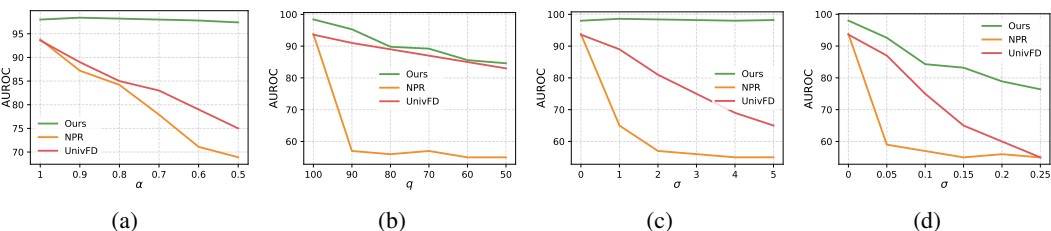

| (a) | (b) | (c) | (d) |

Figure 7: Robustness to perturbations: (a) Rescaling; (b) JPEG compression; (c) Gaussian blur; (d) Gaussian noise.

Table 8: Accuracy comparison with varying patch sizes on the **GenImage** dataset.

| Patch-size | SDv1.4 | SDv1.5 | Midjourney | ADM | GLIDE | Wukong | VQDM | BigGAN | **Mean** |
|---|---|---|---|---|---|---|---|---|---|
| 128 | 99.9 | 99.9 | 92.1 | 94.2 | 98.8 | 99.7 | 99.7 | 99.9 | 98.0 |
| 64 | 99.9 | 99.8 | 92.1 | 89.7 | 99.6 | 99.6 | 99.1 | 99.3 | 97.4 |
| 32 | 99.6 | 99.7 | 91.3 | 87.8 | 97.7 | 99.2 | 98.0 | 99.2 | 96.6 |
| 16 | 95.6 | 95.8 | 88.1 | 84.4 | 88.8 | 93.8 | 89.8 | 92.4 | 91.1 |
| 8 | 89.7 | 91.3 | 83.6 | 73.9 | 83.2 | 86.7 | 78.7 | 79.8 | 83.4 |

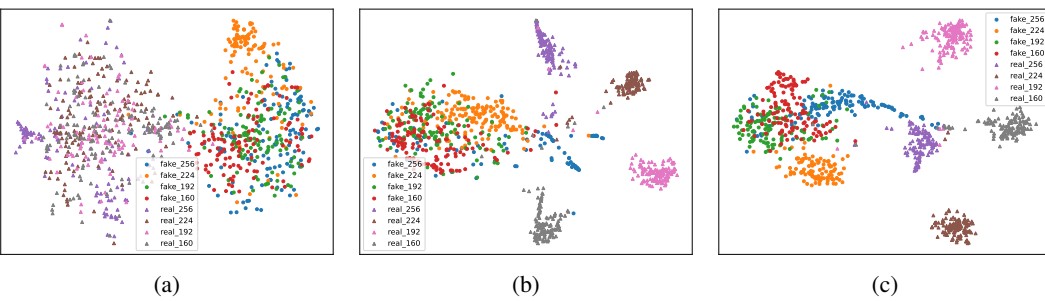

(a)             (b)             (c)

Figure 8: Feature classification performance between generated and real images under different rescaling post-processings: (a) ProGAN; (b) ADM; (c) SDv1.4.

conducted with smaller patch sizes. As shown in the results, the detection performance remains relatively stable when the patch size is reduced to 64 or 32. However, further reduction in patch size leads to a rapid decline in detection effectiveness. This is primarily because smaller patches contain limited rescaling distribution information, making it difficult to extract features that characterize the approximated distribution of generated images. Overall, the pre-trained model demonstrates robustness to variations in patch size, maintaining remarkable detection performance across different configurations.

**t-SNE Visualizations under Mixed Post-rescaling.** Figure 8 presents a unified t-SNE visualization incorporating all post-rescaling images from Figure 6. The three subfigures display generated images and their real counterparts from ProGAN, ADM, and SDv1.4 models, respectively. Each image group comprises the original images along with three variants processed with different rescaling factors. The visualization reveals that real images maintain coherent clustering across various rescaling factors while exhibiting clear separation from rescaled generated images. Furthermore, generated images form distinct clusters corresponding to their respective rescaling factors. Although different rescaling factors differentially affect the approximated rescaling distribution of synthetic images, they fail to obscure the inherent distributional discrepancies. Consequently, the pre-trained model effectively discriminates between post-rescaling generated images and real images. The robustness experiment in Figure 7(a) further confirms our method's resilience to rescaling-based post-processing perturbations.

**Visualization of Periodicity in Rescaled Images.** Rescaled images exhibit distinct periodic distribution patterns, which can be observed through cosine similarity maps derived from features extracted by the pre-trained model. As shown in Figure 9, we present cosine similarity maps of three randomly selected real images after bilinear rescaling with different factors. For clearer visualization, we magnify the top-left 32×32 region to highlight these characteristics. Consistent with our theoretical derivation in the preliminary section, the minimal distribution period of rescaling corresponds to the denominator of the rescaling factor's reduced fractional form. In subfigure (a), where 256/224 simplifies to 8/7, the minimal period is 7, and bright stripes indeed appear at 7-pixel intervals. Similarly, subfigures (b) and (c) demonstrate periods of 3 and 5, respectively. These observations validate our theoretical analysis regarding the periodic distribution properties of bilinear interpolation and confirm the pre-trained model's capability to effectively capture such periodic features.

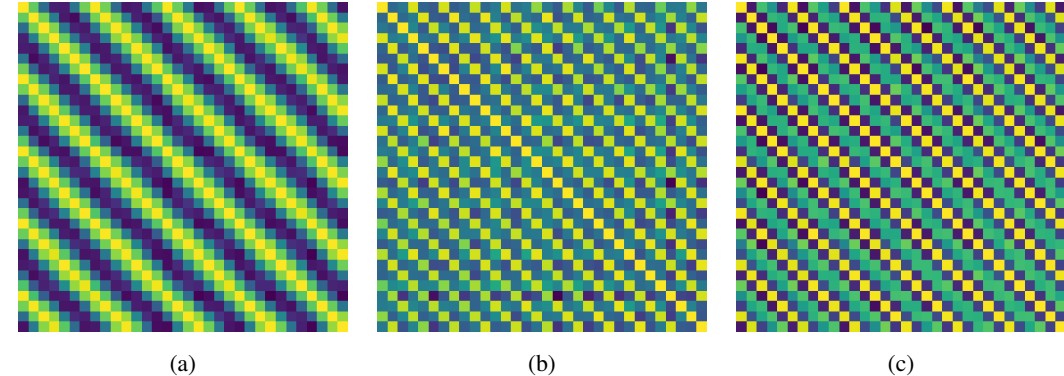

Figure 9: Cosine similarity maps of single images under different rescaling ratios, exhibiting distinct periodicity. (a) Rescaling from 256×256 to 224×224; (b) Rescaling from 256×256 to 192×192; (c) Rescaling from 256×256 to 160×160.

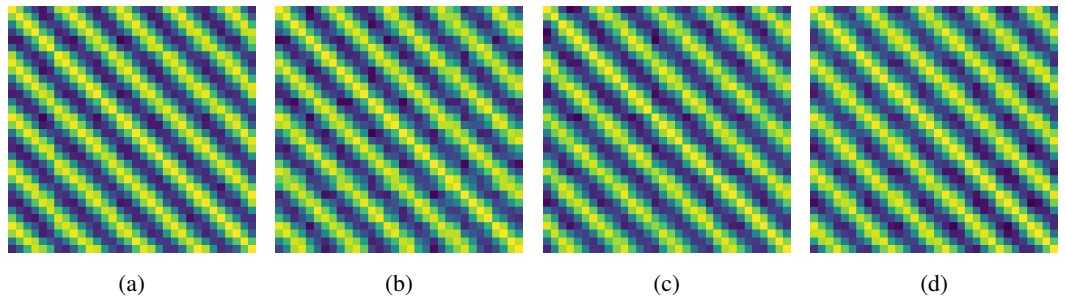

Figure 10: Consistent periodicity across interpolation methods, compatible with BILINEAR-pretrained feature extraction. Rescaling 256×256 to 224×224 via: (a) BOX; (b) BICUBIC; (c) LANCZOS; (d) HAMMING interpolation.

**Consistent Periodic Distribution Across Interpolation Methods.** We employed various interpolation methods to process images and computed their corresponding cosine similarity maps. As shown in Figure 10, we present results from several PyTorch-provided interpolation approaches: BOX, BICUBIC, LANCZOS, and HAMMING. Although the pre-trained model was exclusively trained on bilinear interpolation distributions, it successfully captures periodic features from other interpolation techniques. This observation aligns with our theoretical analysis that different interpolation methods share similar periodic distribution characteristics. Furthermore, these results validate our model's generalization capability across diverse interpolation schemes.

