# OpenReview forum: "Generalization through Discrepancy: Leveraging Distributional Fitting Gaps for AI-Generated Image Detection"
_ICLR.cc/2026/Conference — Submitted to ICLR 2026_

### Official Review · Reviewer_38mx · 2025-10-28

**Soundness:** 2
**Presentation:** 3
**Contribution:** 2
**Rating:** 4
**Confidence:** 4

**Summary:**

The paper aims to improve the generalization ability in the deepfake detection task under the cross-generator scenario, e.g., detection inference with unseen generators from training. Specifically, the authors hypothesize that broadening the coverage of prior-based assumptions would help with the generalization performance and propose to rescale the distribution of the discrepancy to tackle this. Methodologically, the paper presents an analysis of the rescaling properties via bilinear interpolation and leverages a distributional contrastive pre-training mechanism before fine-tuning the classifier. Experiments are conducted on several benchmarks with different image generators.

**Strengths:**

- The paper has a rather clear structure and presents a generally good literature review in terms of existing deepfake detection methods.

- The perspective to address the generalization ability via the distributional discrepancy is somewhat novel and reasonable.

**Weaknesses:**

While the paper presents a relatively complete structure, several important details are still missing to fully evaluate the proposed method and validate its effectiveness.

1. Motivation clarity: I understand that different generators model data distributions differently and may introduce distributional discrepancies during the generation process. However, the connection between the bilinear interpolation-based scaling behavior (Section 3) and its effectiveness for deepfake detection remains unclear. A more concrete justification or intuition for why this approach works would be helpful.

2. Missing relevant baseline: The paper seems to overlook some recent relevant work—such as [a]—which similarly explores disentangling semantic features to improve cross-generator generalization. A discussion and comparison with such baselines would strengthen the positioning of the proposed method.

3. Interpolation choices: The authors state that “we focus on bilinear interpolation to elucidate the underlying mechanisms, noting that other interpolation techniques exhibit analogous properties.” This claim is not self-evident and would benefit from empirical support. Including ablations on alternative scaling methods could help validate this assumption.

4. Additional ablations needed: Some other important ablations appear to be missing and are necessary to support the key claims of the paper. Please refer to the Questions section for specific suggestions.

---
[a] D3: Scaling Up Deepfake Detection by Learning from Discrepancy. In CVPR 2025.

**Questions:**

1. The authors mention the computing resources used in the experiments, but the actual GPU hours are missing. Could the authors clarify the training time required for both the pre-training and fine-tuning stages?

2. Does the patch size used during the pair construction stage impact the detection performance? If so, it would be helpful to provide an ablation or brief discussion on its effect.

3. Additionally, could the authors further clarify the motivation behind the positive and negative sample mechanism illustrated in Figure 4? Specifically, is the rescaling stage applied only to real images, or does it also affect synthetic ones?

---

> ### Author Response · Authors · 2025-12-02
> **Response to Reviewer 38mx (1/2)**
>
> We sincerely thank the reviewer for the thorough evaluation and the recognition of our work's novel and reasonable perspective. We have carefully addressed the concerns regarding the methodological motivation, baseline comparisons, and experimental completeness by providing further theoretical justification and conducting extensive additional experiments (detailed in the Appendix).
>
> ## Response to W1: Justification for Bilinear Interpolation as the Core Assumption
>
> The rescaling operation is an inherent and necessary preprocessing step in the training of generative models. Consequently, the statistical properties of the rescaling distribution are learned and approximated by these models.
> - **Theoretical Foundation (Sec. 3 & App. A.2):** We provide formal derivations showing that interpolation-based rescaling (including bilinear) yields distributions with periodicity and local dependency characteristics.
> - **Detection Mechanism:** While generative models approximate these distributions, they cannot perfectly replicate the full diversity of precise mathematical rescaling from source data. This results in "approximated rescaling distributions" that deviate from precise rescaling. Our fine-grained contrastive learning on bilinear interpolation allows the classifier to detect these fine-grained approximations in generated images.
> - **Generalization Evidence:** Mathematical analysis (App. A.2) and empirical results (Figure 10) confirm that a model trained on bilinear interpolation generalizes to detect images rescaled with other interpolation methods, supporting our choice of bilinear as a representative basis.
>
> ## Response to W2: Comparative Analysis with Paper [a] and Other Baselines
>
> - **Methodological Distinction:** Our method is rooted in the intrinsic fitting property of generative models, specifically targeting the detection of semantic-agnostic, preprocessing-level distributions (e.g., approximated rescaling patterns). In contrast, Paper [a]'s shuffling-based method cannot fully disentangle semantic information, and its reliance on the CLIP backbone inherently prioritizes high-level semantic features, which may confound the detection objective.
> - **Experimental Superiority & Generalization:** Our approach is evaluated under a more challenging single-model to other models generalization scenario.
>     - **State-of-the-Art Performance:** Despite being fine-tuned on only a single source model, our method achieves superior results: 98.0% average accuracy on **GenImage** and 98.6%/99.6% average precision on GAN/Diffusion models in **UniversalFakeDetect**.
>     - **Direct Comparison with [a]:** Paper [a]'s method, even when fine-tuned on multiple models, achieves 86.7% average accuracy and 94.3% average precision in its generalization tests (Table 1). Its performance degrades significantly to 68% average accuracy (Figure 2) when fine-tuned on a single model, highlighting our method's stronger generalization from limited training data.
>     - **Broader Baselines:** We also compare against other semantic-disentangling approaches (e.g., Effort, NPR in Table 2), further validating the effectiveness of our distribution-focused paradigm.
>
> ## Response to W3: Generalization to Other Interpolation Methods
>
> Our method extends beyond bilinear interpolation, even though the preliminary analysis and pre-training framework presented in the main text primarily center on this technique.
> - **Theoretical Basis (Appendix A.2):** We provide mathematical derivations for various interpolation methods. Among PyTorch's six common methods (**nearest, box, bilinear, bicubic, lanczos, hamming**), all except nearest interpolation exhibit the periodicity and local dependency characteristic of bilinear interpolation. Nearest interpolation is excluded from practical consideration as it is not used in generative model preprocessing due to the severe image distortion it introduces.
> - **Empirical Validation:** Since existing generative models typically default to bilinear for data preprocessing, to rigorously test this, **we trained a new ADM generative model specifically using Bicubic interpolation.** Testing revealed robustness across interpolation methods: our Bilinear-pretrained model detects Bicubic-trained generative models with a high accuracy of **98.5%**, demonstrating that the learned features are transferable across different kernels. This indicates that the "approximation error" learned by our model is a fundamental characteristic of generative networks, transferable across different interpolation kernels. See the expanded cosine similarity maps extracted under bilinear pre-trained models for other interpolation methods in Figure 10, demonstrating their similar periodic distribution characteristics.

---

> ### Author Response · Authors · 2025-12-02
> **Response to Reviewer 38mx (2/2)**
>
> ## Response to Q1: Training Time of the Proposed Method
>
> The GPU hours required are approximately 70 hours for the pre-training stage and about 1 hour for the fine-tuning stage.
>
> ## Response to Q2: Impact of Patch Size
>
> We have supplemented an analysis of the impact of different patch sizes on experimental results, as shown in **Table 8**. The model was trained with a patch size of 128, and tested with smaller patch sizes. When the patch size is 64 or 32, the detection results remain largely unaffected. However, as the patch size decreases further, the limited information of rescaling distributions within smaller patches makes it difficult for the model to distinguish between approximated and authentic distributions, leading to a rapid decline in detection accuracy. Overall, the detection performance remains stable when the patch size is sufficiently large.
>
> ## Response to Q3: Sample Construction Rationale in Pre-training
>
> The pre-training stage (Figure 4) exclusively uses **real images** to learn the fine-grained characteristics of precise interpolation distributions.
> - **Sample Construction Logic (Sec. 3 & App. A.2):** The selection of positive/negative pairs is governed by the periodicity and local dependency of interpolation distributions. Specifically:
>     - **Positive Pairs:** Patches located at periodically equivalent positions are treated as positive samples, as they share identical local interpolation distributions (derived from the same neighborhood weighting).
>     - **Negative Pairs (Intra-image):** Patches from non-periodic positions within the same rescaled image are treated as negative samples, as their local interpolation distributions differ.
>     - **Negative Pairs (Cross-image):** Patches from images rescaled with different factors are always negative, as their interpolation characteristics never align.
> - **Learning Outcome:** This rigorous contrastive scheme forces the model to learn a fine-grained representation of precise interpolation distributions across rescaling factors. This foundational knowledge enables it to later identify the approximated distributions characteristic of generated images during detection.

---

### Official Review · Reviewer_TxtG · 2025-10-31

**Soundness:** 3
**Presentation:** 3
**Contribution:** 2
**Rating:** 4
**Confidence:** 3

**Summary:**

This paper proposes a novel approach for detecting AI-generated images by leveraging the discrepancy between mathematically precise image rescaling traces and the imperfect approximations learned by generative models. The authors introduce a contrastive pre-training framework that sensitizes feature extractors to subtle rescaling artifacts using only real images, focusing on the periodic patterns and position shift properties of rescaling operations. Their method achieves state-of-the-art performance across both GAN and diffusion-generated image benchmarks, demonstrating strong generalization capabilities for cross-model detection tasks.

**Strengths:**

1. The paper is clearly written with a complete structure and is easy to follow.
2. I believe the proposed approach is innovative, particularly the method of constructing positive and negative sample pairs, which effectively demonstrates how the technique focuses on rescaling artifacts.
3. Comparative experiments are extensive and comprehensive, and the results validate the effectiveness of the proposed method.

**Weaknesses:**

1. The paper assumes that generators learn and approximate real-world rescaling artifacts from training data. However, given the diverse rescaling artifacts of training datasets (they scale from different resolutions to the same resolution), generators may not consistently learn a specific rescaling pattern, which means their outputs might lack these artifacts(as supported by Figure 5). This raises my concerns that the method might simply detect whether images contain rescaling artifacts rather than determining if they're generated. (Note that both LSUN and ImageNet are preprocessed real datasets, so they have rescaling artifacts).
2. Based on the above analysis, I think the experiment in Figure 6 lacks rigor. A more reasonable scenario would involve applying varying degrees of multiple rescaling operations to both real and fake images to eliminate potential differences caused by preprocessing.
3. The paper lacks robustness testing against common post-processing operations like Gaussian blur and JPEG compression.

Minor weakness: While generally clear, some sections contain unnecessary redundancy, such as the detailed explanation of bilinear interpolation in Section 3 and the problem definition in Section 4.1, where the authors could simplify the presentation.

**Questions:**

Please see Weaknesses.

---

> ### Author Response · Authors · 2025-12-02
> **Response to Reviewer TxtG (1/1)**
>
> We sincerely appreciate the reviewer's valuable comments and recognition of the innovation in our method. In response, we have addressed the issues concerning the reliance of the proposed method on rescaling artifacts, differences in preprocessing rescaling, and questions regarding robustness (detailed in the Appendix).
>
> ## Response to W1: Robustness to Diverse Approximated Rescaling Distributions
>
> - **Core Claim:** Our model **does not** rely on any specific rescaling pattern but detects the generic trace of “approximated rescaling distributions” produced by generative models.
> - **Mechanism:** Through contrastive pre-training on real data, the model learns fine-grained features of “precise” rescaling distributions.
>     - **Detection Principle:** During testing, the classifier detects various approximated rescaling distributions as out-of-domain instances, indicating that the classification is not based on any specific approximated rescaling distribution pattern but on the **fundamental discrepancy** between “precision” and "approximation”.
> - **Empirical Support:**
>     - **Performance on Trace-Free Real Images:** Supplementary results on the **Synthbuster** dataset (where real images are high-resolution RAISE images devoid of any post-processing traces) in Table 7 demonstrate consistent strong performance. This validates the model's generalization ability and confirms that it does not merely rely on the presence or absence of rescaling artifacts.
>     - **Robustness to Post-Processing:** The robustness tests involving post-rescaling operations in Figure 7 & 8 show that the model’s efficacy remains unaffected by such perturbations. This further proves that the model extracts subtle, approximated distributional characteristics of generated images at a more fundamental level.
>
> ## Response to W2: Robust Feature Separation under Post-Rescaling
>
> The features extracted by our model are robust, maintaining clear separability between real and generated images in the representation space even under complex post-rescaling perturbations.
> - **Feature Visualization (Figure 8):** t-SNE visualizations under mixed post-rescaling scenarios show that real images maintain effective clustering after undergoing rescaling with different factors and remain clearly distinguishable from generated images.
> - **Quantitative Robustness (Figure 7a):** Detection performance remains stable across various post-rescaling operations, quantitatively confirming that the model's effectiveness is not compromised by such perturbations.
>
> ## Response to W3: Robustness to Common Image Perturbations
>
> The proposed detection method demonstrates strong robustness against a range of common image perturbations that could obscure forensic traces.
> - **Results:** Our fine-grained contrastive pre-training compels the model to learn robust structural periodicity, not fragile artifacts. Consequently, it maintains high accuracy against perturbations like Rescaling, JPEG, Blur, and Noise (Figure 7).
> - **Specific to Rescaling:** Crucially, our method detects the intrinsic "approximated" distribution of generated images even after they are post-rescaled (which may destroy specific artifacts), preventing evasion of detection (Figure 7 & 8).

---

### Official Review · Reviewer_MDWE · 2025-10-31

**Soundness:** 3
**Presentation:** 3
**Contribution:** 3
**Rating:** 6
**Confidence:** 4

**Summary:**

This paper improves the generalization of deepfake detectors by exploiting a pre-processing distributional discrepancy between real and generated images. Focusing on the rescaling operation, the authors analyze bilinear interpolation and reveal two key properties—periodicity and local pixel dependency—that differ across real and synthetic data. They introduce an unsupervised contrastive learning framework to model these properties, achieving state-of-the-art results on the Self-Synthesis and GenImage benchmarks

**Strengths:**

The method presented is an unsupervised method which harnesses the distributional properties of a simple yet overly common preprocessing step: rescaling. This approach leads to a good generalization when evaluated on benchmarks such as GenImage or Self-Synthesis.
To further prove the efficiency of their method, the authors conducted an ablation study demonstrating the efficiency (1) of their pre-training strategy (2) of their pre-trained backbone compared to CLIP and (3) of the few-shot fine-tuning scenario. Also, the authors present t-SNE plots of the extracted features (which further supports the second claim of the ablation analysis) and cosine similarity maps of both real and fake images which illustrates the periodicity property of the former as opposed to the latter.

**Weaknesses:**

The authors claim high capabilities in terms of generalization. Although cross-generator performance is tested, a cross family of generators evaluation is missing. (e.g. an analysis on the UniversalFakeDetect Dataset).
The model is pre-trained on ImageNet and then fine-tuned and tested on GenImage benchmark. This is an issue since GenImage contains real samples from ImageNet.
Claims related to robustness against post-scaling operations are not well supported by quantitative results, but only t-SNE plots. The authors should test their model in this setup.
In section 3, the authors claimed that other interpolation methods apart from bilinear interpolation exhibit similar properties. However, this claim is not supported either mathematically or empirically.
In Figure 6 the labels are too small and not clear.

**Questions:**

In section 5.2 “Generated Data Exhibits Distinct Cosine Similarity Map.”: why certain generative models do have multiple bright parallel bands (e.g. GLIDE) while others don’t (e.g. ProGAN)? Also, can one observe the rescaling factors based on the distance between the bright lines? It would be interesting to observe this effect and, if proven empirically, it would further demonstrate the correlation with the periodicity property.

---

> ### Author Response · Authors · 2025-12-02
> **Response to Reviewer MDWE (1/2)**
>
> We sincerely thank the reviewer for the positive assessment and for recognizing the novelty of our unsupervised framework and its effectiveness. We have addressed concerns regarding cross-family generalization, data overlap, and quantitative robustness with extensive new experiments (detailed in the Appendix).
>
> ## Response to W1: Cross-Family Generalization
>
> We have added the **UniversalFakeDetect** dataset to our evaluation (Table 5 & 6). Our model demonstrates strong generalization across both GANs and Diffusion models, achieving mean AP of **98.6% and 99.6%**, which confirms its effectiveness beyond the specific architectures seen during training.
>
> ## Response to W2: Data Leakage Concern
>
> To rule out the benefit from ImageNet pre-training overlap, we tested on datasets with **totally disjoint real image sources**:
> - **UniversalFakeDetect:** Uses **Laion** as real data (Table 5, Table 6).
> - **Synthbuster:** Uses **RAISE** as real data (Table 7).
> - **Self-Synthesis:** Uses **LSUN** as real data (Table 1).
> - **Result:** As shown in Table 1, 5, 6, and 7, our model maintains SOTA performance across all these datasets. This proves that our model learns universal "rescaling discrepancy" rather than memorizing ImageNet semantics.
>
> ## Response to W3: Quantitative Robustness Analysis
>
> Thanks to the reviewer, we have supplemented **comprehensive quantitative results** in Figure 7 and 8.
> - **Quantitative Evaluation:** Following Wang et al. (2020) and Ojha et al. (2023), we evaluated the model under rescaling, JPEG, Gaussian Blur, and Gaussian Noise.
> - **Results:** Specifically for post-rescaling (the reviewer's main concern), our model exhibits remarkable stability. Even under significant rescaling factors (e.g., 1x - 0.5x), the performance drop is negligible (< **3% AUC**), whereas baseline methods fail significantly.
> - **Reasoning:** This supports our claim: while the model learns from rescaling traces, the contrastive objective forces it to learn intrinsic structural discrepancies that persist even after the image is resized again. (See Figure 8 for visualized clusters).
>
> ## Response to W4: Generalization to Other Interpolation Methods
>
> We have formalized the claim regarding other interpolation methods in **Appendix A.2**.
> - **Mathematically:** We provide derivations showing that **Bicubic, Lanczos, Box, and Hamming** interpolations all share the key mathematical properties utilized by bilinear interpolation in our model: **periodic distribution** and **local dependency**. (Suitable for 5 out of 6 standard PyTorch interpolation methods. Nearest Neighbor is the exception, but is rarely used in high-quality generation).
> - **Empirically:** Since existing models typically default to bilinear for data preprocessing, to rigorously test this, **we trained a new ADM generative model specifically using Bicubic interpolation.** Testing revealed robustness across interpolation methods: our Bilinear-pretrained model detects Bicubic-trained generative models with a high accuracy of **98.5%**, demonstrating that the learned features are transferable across different kernels. This indicates that the "approximation error" learned by our model is a fundamental characteristic of generative networks, transferable across different interpolation kernels. See the expanded cosine similarity maps extracted under bilinear pre-trained models for other interpolation methods in Figure 10, demonstrating their similar periodic distribution characteristics.

---

> ### Author Response · Authors · 2025-12-02
> **Response to Reviewer MDWE (2/2)**
>
> ## Response to Q1: Interpreting Differences in Cosine Similarity Maps (GLIDE vs. ProGAN)
>
> The cosine similarity maps exhibit variations across different generative models, which we hypothesize may be influenced by factors such as model architectures, training data distributions, and sampling steps. We will investigate the specific reasons behind these variations in future work. A common characteristic observed is the significant divergence from the distribution of real datasets: real data display denser bright parallel bands, whereas generated data concentrate bright stripes predominantly near the diagonal region (as shown in Fig. 5).
>
>
> ## Response to Q2: Interpreting Rescaling Factors in Cosine Similarity Maps
>
> To further explore the relationship between periodicity characteristic and rescaling factor, we have supplemented cosine similarity maps of images under different rescaling factors, as shown in Figure 9.
> - **Periodicity Distribution:** The periodicity of the interpolation distributions is manifested as **periodic bright band patterns** in the cosine similarity maps, with the spacing between bands determined by the denominator of the simplest fractional form of the rescaling factor. As shown in Figure 9, the periods corresponding to three different rescaling factors strictly adhere to this pattern, which aligns with our theoretical derivation and confirms the sensitivity of pre-trained models to interpolated distributions.
> - **Rescaling Factor:** Different scaling factors may share the same denominator and thus exhibit identical periods. Consequently, direct observation reveals periodic relationships rather than specific rescaling factors. To derive precise rescaling factors from cosine similarity maps, a convolutional model could be further trained to map the value distribution of bright and dark stripes to specific rescaling factors.

---

### Official Review · Reviewer_vBs2 · 2025-11-01

**Soundness:** 3
**Presentation:** 2
**Contribution:** 2
**Rating:** 4
**Confidence:** 4

**Summary:**

This paper indicates the inherent periodic patterns and position shift properties of real images. Accordingly, they propose a fake image detection method based on a contrastive pre-training framework using only real images to learn the discrepancy between mathematically precise image rescaling traces and the imperfect approximations learned by generative models.

**Strengths:**

1. This method reveals the nature of periodic patterns and position shift properties of bilinear interpolation, serving as guidance for the detection method.

2. This paper is well-written.

**Weaknesses:**

1. The proposed method is based on the assumption that real images contain mathematically precise image rescaling traces, while fake images only exhibit imperfect approximations of them. However, real images are not necessarily rescaled, in which case this assumption does not hold. Moreover, this assumption about fake images is not demonstrated in the preliminary analysis. In particular, according to prior studies [A], fake images may still retain preprocessing traces from their source images, including rescaling traces. This weakens the motivation of the paper and raises concerns about the validity of its underlying premise.

2. The proposed method is designed based on rescaling, which is a low-level feature. Such low-level features are generally sensitive to post-processing perturbations. In particular, prior research [A] has demonstrated that post-processing operations can interfere with each other and even destroy forgery artifacts in fake images. Therefore, the robustness of the proposed method needs to be thoroughly evaluated.

3. According to prior research [B], the training data used in the model contain biased image formats. Therefore, it is necessary to evaluate the performance of models on unbiased image formats, such as the test data used in [B] or the SynthBuster [C] dataset.

4. Since both the preliminary analysis and the proposed method are based on the properties of bilinear interpolation, it is necessary to evaluate whether the model remains effective when the test data are rescaled using other interpolation methods.

5. In Figure 6, the t-SNE visualizations of the original images and the post-rescaling images are plotted separately. These samples should be visualized together to show whether the post-rescaled samples are misclassified within the original feature space. Alternatively, experimental results should be provided to demonstrate whether fake images become misclassified as real after post-rescaling.

[A] Corvi, Riccardo, et al. "Intriguing properties of synthetic images: from generative adversarial networks to diffusion models." Proceedings of the IEEE/CVF conference on computer vision and pattern recognition. 2023.

[B] Grommelt, Patrick, et al. "Fake or jpeg? revealing common biases in generated image detection datasets." European Conference on Computer Vision. Cham: Springer Nature Switzerland, 2024.

[C] Bammey, Quentin. "Synthbuster: Towards detection of diffusion model-generated images." IEEE Open Journal of Signal Processing 5 (2023): 1-9.

**Questions:**

see weaknesses

---

> ### Author Response · Authors · 2025-12-02
> **Response to Reviewer vBs2 (1/2)**
>
> We sincerely thank the reviewer for the insightful comments and the recognition of our method’s novelty regarding periodic patterns. We have carefully addressed the concerns regarding the underlying assumption, robustness, and dataset bias by conducting extensive additional experiments (detailed in the Appendix).
>
> ## Response to W1: Validity of the Core Assumption
>
> - **W1.1 Clarification on "Real" vs. "Fake" Traces:** We clarify that our premise **does not** require test-time real images to contain rescaling traces.
>     - **Generative Trace:** The "imperfect approximations" in fake images arise because generative models attempt to learn the distribution of their training data (which contains rescaling traces because the "Rescaling" operation always shows in the pre-processing steps of generative models, which does not matter whether the images in the original datasets are rescaled). Consequently, generated images inherently carry an _approximated_ version of these traces.
>     - **Real Images:** Real images fall into two categories: (1) those with mathematically precise rescaling traces (if post-processed), or (2) pristine images with no such traces (e.g., RAISE).
>     - **Evidence:** Our model is designed to distinguish "approximated traces" (fake) from "precise or absent traces" (real). Thanks to the reviewer, we present new results about "absent traces" on **Synthbuster** (Table 7). Our model distinguishes generated images from high-resolution, unprocessed RAISE images, **achieving an average AUC of 97.0%**. This quantitative evidence confirms that our model does not falsely rely on real images having a rescaling history.
>
> - **W1.2 Assumption Visualization:** We demonstrated the assumption about the "approximated traces" (fake) from "precise or absent traces" (real) by cosine similarity maps in Figure 5 (Expe in original paper). The bright stripes observed in real images indicate that they comprise diverse rescaling distributions, whereas generated images exhibit sparse bright stripes. Moreover, different generative models demonstrate similar characteristic distributions, revealing the averaged interpolation patterns that emerge after generative models approximate complex rescaling distributions. This analysis requires prior introduction of the pre-trained models to be fully comprehensible. Therefore, we deliberately placed this discussion in the experimental section rather than in the preliminary analysis to ensure proper contextual understanding.
>
> - **W1.3 Consistency with Prior Work [A]:** Our work is actually inspired by and consistent with [A]. While [A] states that fake images retain preprocessing traces, our key insight is that these retained traces are statistically distinguishable approximations, not mathematically perfect replicas. Our method explicitly learns this subtle discrepancy. Our core assumption is not that preprocessing traces are absent in generated images. This does not weaken our motivation but is the core motivation that enables our superior performance.
>
> ## Response to W2: Robustness against Post-processing Perturbations
>
> We have supplemented comprehensive robustness experiments against post-processing in the new **Figure 7 and Figure 8 (Appendix)**.
>
> - **Results:** The fine-grained contrastive learning during pre-training forces the model to capture robust structural periodicity rather than fragile artifacts. As shown in Figure 7, our method maintains high accuracy under Rescaling, JPEG compression, Gaussian blur, and Gaussian noise.
> - **Specific to Rescaling:** Even when fake images are re-rescaled (which might destroy artifacts), our model successfully detects the underlying "approximated" distribution features, preventing the evasion of detection (see Figure 7 & 8).
>
> ## Response to W3: Evaluation on Unbiased Datasets
>
> We have added evaluations on **GenImage (unbiased)** and **Synthbuster** (Table 4 & 7).
>
> - **GenImage (Unbiased):** In this dataset, generated images undergo JPEG compression same to real images, removing compression format shortcuts. Our method achieves **98.4% mean Accuracy**, surpassing the second best SOTA method B-Free(Guillaro et al., 2025) by **+9.1%**, proving it does not rely on simple JPEG/PNG cues.
> - **Synthbuster:** We tested high-quality diffusion generative models and high-resolution real images without post-processing, obtaining **97.0% AUC**. Since the real images here have no post-processing history, this further validates that our model **detects the presence of generative artifacts** rather than the absence of real-world processing traces.

---

> ### Author Response · Authors · 2025-12-02
> **Response to Reviewer vBs2 (2/2)**
>
> ## Response to W4: Generalization to Other Interpolation Methods
>
> Our method is not limited to bilinear interpolation, although our preliminary analysis and pre-training discussions primarily focused on it.
>
> - **Theoretical Basis:** As detailed in **Appendix A.2**, 5 out of 6 standard PyTorch interpolation methods (box, bilinear, bicubic, lanczos, and hamming, except 'nearest') share the key properties of periodic distribution and local dependency utilized by our model. While nearest interpolation lacks these characteristics, it is not employed in generative model data preprocessing due to the severe image distortion it causes during rescaling.
> - **Empirical Validation:** Since existing generative models typically default to bilinear for data preprocessing, to rigorously test this, **we trained a new ADM generative model specifically using Bicubic interpolation**. Testing revealed robustness across interpolation methods: our Bilinear-pretrained model detects Bicubic-trained generative models with a high accuracy of **98.5%**, demonstrating that the learned features are transferable across different kernels. This indicates that the "approximation error" learned by our model is a fundamental characteristic of generative networks, transferable across different interpolation kernels. See the expanded cosine similarity maps extracted under bilinear pre-trained models for other interpolation methods in Figure 10, demonstrating their similar periodic distribution characteristics.
>
> ## Response to W5: Visualization of Misclassification Risks
>
> We have updated the t-SNE visualizations (original Figure 6) in **Figure 8** to plot original and post-rescaled samples together as suggested by the reviewer. The visualization clearly shows that post-rescaled fake images do not drift into the cluster of real images. They remain separable in the feature space, aligning with the robust quantitative results reported in Figure 7(a) in Response W2.

---

### Author Response · Authors · 2025-12-03
**Summary for Reviewers and AC (2/2)**

### **3. Generalization and Evaluation on Unbiased/Disjoint Datasets**

Reviewers requested evaluations on unbiased datasets to rule out format-based shortcuts (JPEG and PNG for fake and real), called for broader cross-family validation (e.g. UFD), and raised concerns about potential data leakage arising from the overlap between ImageNet pre-training and test data.

**Evaluation on Unbiased Datasets (vBs2-W3):**

- **GenImage(Unbiased):** Generated images undergo JPEG compression, the same as real images, removing compression format shortcuts. Achieved **98.4%** mean Accuracy, surpassing B-Free by **+9.1%,** proving it doesn't rely on compression format cues.
- **Synthbuster:** Tested on high-quality diffusion models vs. pristine RAISE images (no processing traces), achieving **97.0%** AUC. This confirms detection of _generative artifacts_, not the absence of real-world traces.

- **Cross-Family Generalization (MDWE-W1):** Evaluated on **UniversalFakeDetect**, achieving mean AP of **98.6%** (GANs) and **99.6%** (Diffusion), demonstrating strong generalization.

- **Ruling Out Data Leakage (MDWE-W2):** We tested on datasets with disjoint real image sources: UniversalFakeDetect (**LAION**), Synthbuster (**RAISE**), and Self-Synthesis (**LSUN**). SOTA performance across all proves learning of universal "rescaling discrepancy," not ImageNet semantics.

### **4. Methodological Details and Analysis**

Reviewers requested deeper interpretations of the observed cosine similarity map patterns and asked for clarifications on implementation details such as computational cost, patch size sensitivity, and pre-training sample construction.

### Visualization & Interpretation (vBs2-Q1, vBs2-Q2):

- **Map Differences:** Acknowledged variations may stem from architecture/data/steps; commonality is denser bands in reals vs. diagonal concentration in fakes.
- **Rescaling Factors:** Added Figure 9 showing band period correlates with rescaling factor denominator. Precise factor estimation would require an additional mapping model.

- **Implementation Details (38mx-Q1, 38mx-Q2, 38mx-Q3):**
  - **Training Time:** 70 GPU hours (pre-train) + 1 hour (fine-tune).
  - **Patch Size:** Analysis in Table 8 shows **stable performance** for various patch sizes.
  - **Pre-training Logic:** Uses **only real images**. Positive/negative pairs are defined by the periodicity and local dependency of precise interpolation distributions to learn fine-grained representations.

We believe these substantial updates and additional results reinforce the soundness and robustness of our proposed framework. We remain at your disposal for any further clarifications.

Best regards,

The authors

---

### Author Response · Authors · 2025-12-03
**Summary for Reviewers and AC (1/2)**

Dear Area Chair,

We sincerely appreciate the time and effort dedicated by the reviewers and the Area Chair to our submission. We understand the immense workload involved in coordinating the review process. To facilitate your final assessment, we provide a concise summary of our work, the key concerns raised by reviewers, and how we have addressed them below by conducting additional experiments, providing necessary visualizations, and responding point-by-point in our rebuttal.

## Paper Overview

We propose a generative image detection method by exploiting the **discrepancy** between precise and approximated rescaling distributions, which arise from generative models' inability to fully capture the **periodicity and local dependency** in rescaling distributions. Through fine-grained contrastive learning on real images, our model learns to detect these subtle distributional differences. It achieves **98.0%** average accuracy on GenImage (+2.2% over SOTA) and **97.2%** on Self-Synthesis (+4.0% over SOTA). Extensive theoretical and robustness analyses validate our claims. We hope our work can offer valuable insights and a new perspective to the community, inspiring further advancements in this field of research.

## Reviewer Recognition

The reviewers generally recognized the novelty and effectiveness of our proposed method and the presentation of our paper:

- Reviewers **vBs2** and **MDWE** highlighted the importance of our discovery regarding the **periodicity and local dependency characteristics** within rescaling distributions, noting that it provides **crucial guidance** for the proposed method.
- Reviewers **MDWE** and **TxtG** further recognized the innovative nature of the fine-grained contrastive learning task we designed based on rescaling distributions.
- Reviewer **38mx** acknowledged the novelty and reasonableness of our approach, which detects fake images by exploiting distributional discrepancies inherent to generative models.
- Reviewers acknowledged the method achieves "state-of-the-art results" (MDWE, TxtG) and demonstrates "good generalization" (vBs2). All reviewers agreed that the paper was well-written and structured.

## Key Concerns and Our Responses

### **1. Validity and Generality of the Core Assumption**

Key concerns centered on whether our assumption holds for real images without rescaling operation, whether the method generalizes to non-bilinear interpolations, and how it reconciles with conflicting prior findings [A].

- **Clarification of the Assumption (vBs2-W1):** We explicitly state our premise **does not** require real images to have rescaling traces. Real images are categorized as having either precise traces (if post-processed) or no traces (pristine). The classifier is designed to distinguish *approximated* traces (fake) from *precise or absent* traces (real).

- **Theoretical Justification (38mx-W1):** We provide formal derivations in Appendix A.2 showing that key properties (periodicity, local dependency) are shared by 5/6 common interpolation kernels (excluding 'nearest' which causes severe artifacts and is not applicable in data preprocessing). We empirically validate generality by training a new ADM model with **Bicubic interpolation;** our bilinear-pretrained model detects it with **98.5% accuracy**, proving transferability.

- **Consistency with Prior Work (vBs2-W1.3):** Our work is inspired by [A] but provides a key insight: generative models produce **statistically distinguishable approximations** of preprocessing traces, not **perfect replicas**. Our method explicitly learns this discrepancy, which is the source of its superior performance.

> [A] Corvi, Riccardo, et al. "Intriguing properties of synthetic images: from generative adversarial networks to diffusion models." Proceedings of the IEEE/CVF conference on computer vision and pattern recognition. 2023.

### **2. Robustness to Post-Processing and Perturbations**

Reviewers noted that low-level features might be fragile under perturbations like blurring or re-rescaling, and emphasized that the initial reliance on qualitative t-SNE plots was insufficient to prove robustness.

- **General Robustness (vBs2-W2, TxtG-W3):** We added new experiments in Figure 7 & 8 (Appendix). Our method maintains high accuracy under Rescaling, JPEG, Gaussian Blur, and Noise.
- **Specific Robustness to Re-rescaling (vBs2-W2, TxtG-W2, MDWE-W3):** The fine-grained contrastive pre-training forces the model to learn the **underlying structural periodicity** of the "approximated distribution," which persists even after post-rescaling alters superficial artifacts. t-SNE visualizations (Figure 8) show that re-rescaled fakes remain separable from re-rescaled reals. Quantitatively, performance drop is minimal (e.g., <3% AUC for significant rescaling).

---

### Meta-Review · Area_Chair_UKuk · 2026-01-07

**Summary:**

Across four reviews, the submission is seen as a well-written and potentially novel detector leveraging periodic/local-dependency properties of rescaling distributions via contrastive pre-training. However, the reviews converge on multiple decision-critical concerns: the central “precise vs approximated rescaling traces” premise is insufficiently established for broad real-world conditions (e.g., pristine real images without rescaling history and synthetic images that may inherit preprocessing traces), the method’s reliance on low-level rescaling cues raises fragility risks under common post-processing, and the evaluation protocol risks confounds from dataset bias (format shortcuts) and overlap with ImageNet pretraining. Additional gaps include limited support for claims beyond bilinear interpolation, missing cross-family validation in the original submission, and missing experimental details/ablations needed to fully assess the method. While the rebuttal adds experiments and clarifications, the paper still reads as over-claiming generality/robustness relative to what is convincingly established in the core submission.

**Reviewer Concerns:**

Partially addressed concerns:
- The authors clarify the intended real-vs-fake distinction (approximated traces vs precise/absent traces) and add a trace-free real dataset result. This helps, but it does not fully resolve the underlying conceptual risk noted by reviewers: the mechanism may still correlate with the presence/structure of preprocessing traces rather than “generatedness,” and the rebuttal does not fully establish that the learned cues are stable across diverse real acquisition pipelines and editing histories.
- Added quantitative tests improve the evidence, but given the low-level nature of the signal, reviewers’ concern about interference between operations remains salient; the rebuttal evidence is limited to a set of perturbations and does not fully de-risk adaptive or compound post-processing scenarios.
- The authors add theory and a targeted empirical check, but the breadth of real-world interpolation/preprocessing diversity is not convincingly covered; the argument still hinges on a narrow slice of possible pipelines.

Remaining concerns:
- The rebuttal adds “unbiased/disjoint” evaluations, yet the original concern remains that benchmark construction and shared data sources can unintentionally encode shortcuts; without reviewer follow-up, it is unclear that the new experiments fully rule out dataset-specific cues.
- The link from interpolation analysis to reliable detection remains insufficiently intuitive in the main narrative; the rebuttal helps but does not guarantee the paper itself is now self-contained and persuasive.
- Even with additional results, the overall framing still appears stronger than what is robustly supported, particularly for deployment-like conditions.

**Reviewer Scores:**

- vBs2: likely 4 → 4 (at most 4→5, but concerns about premise/real-world validity remain).
- TxtG: likely 4 → 4 (at most 4→5; still worried about detecting preprocessing artifacts).
- 38mx: likely 4 → 4 (missing details addressed, but motivation/positioning concerns persist).
- MDWE: likely 6 → 6 (may stay mildly positive, but not enough to offset three borderline-below reviews).

---

### Decision · Program_Chairs · 2026-01-26

Reject